# Language-guided Open-world Video Anomaly Detection under Weak Supervision

**Zihao Liu, Xiaoyu Wu**[*]**, Jianqin Wu, Xuxu Wang & Linlin Yang**
State Key Laboratory of Media Convergence and Communication
Communication University of China
{liuzihao, wuxiaoyu}@cuc.edu.cn
{wujianqin, wangxuxu}@mails.cuc.edu.cn mu4yang@gmail.com

## Abstract

Video anomaly detection (VAD) aims to detect anomalies that deviate from what is expected. In open-world scenarios, the expected events may change as requirements change. For example, not wearing a mask may be considered abnormal during a flu outbreak but normal otherwise. However, existing methods assume that the definition of anomalies is invariable, and thus are not applicable to the open world. To address this, we propose a novel open-world VAD paradigm with variable definitions, allowing guided detection through user-provided natural language at inference time. This paradigm necessitates establishing a robust mapping from video and textual definition to anomaly scores. Therefore, we propose LaGoVAD (**La**nguage-**g**uided **O**pen-world **V**ideo **A**nomaly **D**etector), a model that dynamically adapts anomaly definitions under weak supervision with two regularization strategies: diversifying the relative durations of anomalies via dynamic video synthesis, and enhancing feature robustness through contrastive learning with negative mining. Training such adaptable models requires diverse anomaly definitions, but existing datasets typically provide labels without semantic descriptions. To bridge this gap, we collect PreVAD (**Pre**-training **V**ideo **A**nomaly **D**ataset), the largest and most diverse video anomaly dataset to date, featuring 35,279 annotated videos with multi-level category labels and descriptions that explicitly define anomalies. Zero-shot experiments on seven datasets demonstrate LaGoVAD's SOTA performance. Our dataset and code are released at https://github.com/Kamino666/LaGoVAD-PreVAD.

## 1 Introduction

Video Anomaly Detection (VAD) aims to identify frames in videos that deviate from expected patterns (Chandrakala et al., 2023; Wu et al., 2024a), which is applicable in fields such as intelligent surveillance (Pang et al., 2020). In recent years, many VAD methods have achieved commendable performance employing weak supervision (Pu et al., 2024; Joo et al., 2023; Yang et al., 2024b; Chen et al., 2023; Sultani et al., 2018; Wu et al., 2024c) in the closed-set setting. However, there is a consensus (Wu et al., 2024b; Pang et al., 2020; Zhu et al., 2022; 2024b) that the field is moving towards enabling models to detect anomalies beyond the training data in open-world scenarios.

As shown in Fig. 1a, the training data of VAD models encompass patterns labeled as *normal* or *abnormal*, where normal patterns include activities such as running or driving and abnormal patterns include events like explosions. Conventional closed-set methods (Fig. 1b) (Pu et al., 2024; Wu et al., 2024c) aim to detect patterns identical to those encountered during training when applied to test sets, thereby restricting their application in open-world scenarios. In contrast, open-set approaches (Fig. 1c) (Zhu et al., 2022) (including open-vocabulary (Wu et al., 2024b) and domain generalization (Aich et al., 2023; Jain et al., 2024; Wang et al., 2024c) methods) are able to detect novel patterns absent from the training data without tuning. However, these methods neglect the critical issue of potential label change during testing (Fig.1d), i.e., patterns originally labeled as normal may be redefined as abnormal (and vice versa). A representative example from Fig. 1e demonstrates

---

[*]Corresponding author

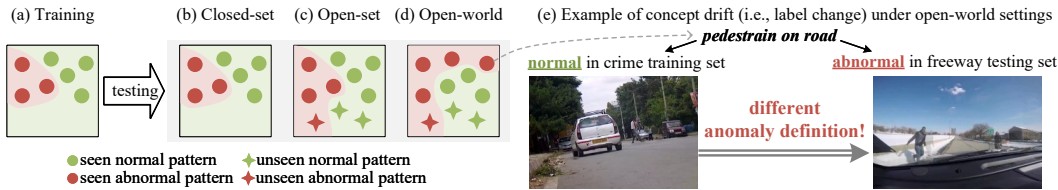

Figure 1: Comparison of different VAD paradigms. Closed-set methods (b) can only detect anomalies in the training scope, while open-set methods (c) can detect novel anomalies. Our open-world approach (d) can deal with label change in open-world scenarios, with an example in (e).

this phenomenon: while *pedestrian on road* is regarded as a normal behavior in conventional crime anomaly datasets, this same pattern would typically be classified as abnormal in freeway surveillance scenarios. The cause of such label change lies in the user's different definition of what constitutes anomalies, driven by environments or temporal policies. Formally, this is a concept drift issue, as defined in (Moreno-Torres et al., 2012), which refers to the divergence between the conditional probability distributions of training and testing phases, i.e., $P_{\text{train}}(Y|V) \neq P_{\text{test}}(Y|V)$, where $V$ are videos and $Y$ are anomaly labels. While some attempts have begun to address this, critical limitations remain. Scene-dependent methods (Cao et al., 2023; Cho et al., 2023; Aich et al., 2023) associate the anomaly definition with scenes, neglecting user-specific requirements (e.g., hospital administrators may require detecting the anomaly of not wearing masks during influenza outbreaks but not at other times). Meanwhile, a dataset-dependent method (Cho et al., 2024) explores anomaly conflicts across datasets, but remains constrained by predefined categories of training datasets. Besides the limitation of task settings, existing methods are evaluated on limited scenes with small-scale data, lacking extensive zero-shot cross-domain comparisons to verify the open-world capability.

To address the concept drift challenge, we propose a novel open-world paradigm. First, we explicitly model the anomaly definition as a stochastic variable instead of fixing it as one or a few realizations. Then, we condition predictions $Y$ on both the video $v$ and the anomaly definition $Z$, i.e., learning a mapping $\Phi : (V, Z) \rightarrow Y$. Since we take the changing definition into account, we effectively avoid concept drift (as detailed in Sec. 3). Finally, to enable natural interaction, we employ textual anomaly definition, allowing users to dynamically define anomalies via language.

However, learning $\Phi$ requires modeling a more complex multimodal space, resulting in decaying sample density that leads to overfitting. To address this, we mitigate it from both model and dataset perspectives. **As for the model**, we propose a **La**nguage-**G**uided **O**pen-world **V**ideo **A**nomaly **D**etector (LaGoVAD), which employs two regularization strategies to reduce overfitting: 1) Aligning vision and language through contrastive learning with negative sample mining, which increases the sample's quality and learns more robust features. 2) Incorporating a dynamic video synthesis module that generates long videos and pseudo-labels on the fly, which diversifys the relative duration of abnormal events. **As for the dataset**, we construct a large-scale diversified **Pre**-training **V**ideo **A**nomaly **D**ataset (PreVAD), collected through a scalable data curation pipeline utilizing foundation models to automate data cleaning and annotation, significantly reducing manual labeling costs while ensuring high quality. PreVAD comprises 35,279 videos annotated with multi-level categories and anomaly descriptions, supporting weakly-supervised training. To our knowledge, PreVAD surpasses existing datasets in diversity and scale.

The model is evaluated under two zero-shot evaluation protocols: one comprehensively assesses open-world capability by evaluating cross-domain performance across seven diverse datasets, addressing key open-world challenges such as detecting unseen categories and handling concept drift, while the other specifically measures concept drift by averaging performance on a dataset under different anomaly definitions. Our contributions are summarized as follows:

1. We reformulate open-world VAD that pioneers the formulation of the concept drift in VAD and proposes a joint modeling paradigm to avoid it.

2. We propose a novel language-guided video anomaly detection model, LaGoVAD, which implements the proposed paradigm and incorporates two regularization strategies to mitigate overfitting.

Table 1: Comparisons between PreVAD and existing datasets. Our dataset 1) has the largest scale and broadest domain coverage, 2) is annotated with abnormal video descriptions, 3) enables zero-shot evaluation without relying on existing VAD datasets.

| Dataset | # videos (# abnormal videos) | Domain | # categories | Text Anno. | Source |
|---|---|---|---|---|---|
| ShanghaiTech (Zhong et al., 2019) | 437 (107) | campus | 13 | - | recording |
| UCF-Crime (Sultani et al., 2018) | 1900 (950) | crime | 14 | - | web |
| XD-Violence (Wu et al., 2020) | 4754 (2405) | crime | 7 | - | web,movie |
| LAD (Wan et al., 2021) | 2000 (762) | crime, traffic, animal, mishap | 14 | - | web |
| TAD (Lv et al., 2021) | 500 (250) | crime | 8 | - | web |
| UBNormal (Acsintoae et al., 2022) | 543 (278) | pedestrian | 28 | - | synthesis |
| DoTA (Yao et al., 2022) | 5677 (5677) | traffic | 9 | - | web |
| MSAD (Zhu et al., 2024b) | 720 (240) | crime, traffic, mishap | 55 | - | web |
| UCCD (Zhou et al., 2024) | 1012 (382) | crime | - | dense | UCF |
| UCA (Yuan et al., 2024) | 1854 (944) | crime | - | dense | UCF |
| VAD-Instruct50k (Zhang et al., 2024) | 5547 (2715) | crime | - | instruction | UCF+XD |
| HAWK (Tang et al., 2024) | 7852 (6677) | crime, traffic | - | instruction | 7 VAD datasets |
| CUVA (Du et al., 2024) | 1000 (1000) | crime, traffic, pedestrian, animal | 42 | instruction | web |
| **PreVAD** | **35279 (11979)** | **crime, traffic, animal, mishap, production** | 35 | anomaly description | web |

3. We build a large-scale and diverse dataset, PreVAD, annotated with multi-level taxonomy and anomaly descriptions to enhance generalization under the new paradigm.

4. We conduct zero-shot cross-dataset evaluation and concept drift evaluation to validate the generalization, where LaGoVAD achieves state-of-the-art performance.

# 2 RELATED WORK

## 2.1 VIDEO ANOMALY DATASETS

We summarize the characteristics of existing video anomaly datasets in Tab. 1. **Scale**: The largest standalone dataset (Wu et al., 2020) contains only 5K videos, with ensemble datasets reaching 7.8K (Tang et al., 2024). The data scarcity limits the performance of VAD. **Domain & Category**: Many datasets focus only on a single scene, such as traffic or campus. The few datasets that cover multiple scenes overlook domains like mishaps, animal-related violence, and production accidents. **Text Annotation**: Existing VAD datasets are labeled with anomaly categories, which introduces semantic ambiguity. Although some datasets provide different types of text annotation, they focus on understanding or captioning tasks and cannot provide a fine-grained overall description of the anomaly in a video. **Source**: Current datasets are mainly from public web videos, while others rely on synthetic generation (Acsintoae et al., 2022; Narwade et al., 2024) or movie clips (Wu et al., 2020). However, synthetic datasets suffer from misalignment with the real world, and movie data raises concerns about potential copyright infringement. In this paper, we propose a scalable data curation pipeline to collect a novel dataset, which has large-scale diversified videos with multi-level taxonomy and anomaly descriptions.

## 2.2 OPEN-WORLD VIDEO ANOMALY DETECTION METHODS

Intuitively, open-world VAD models should detect novel anomalies beyond the training set (Zhu et al., 2022; Wu et al., 2024b; Tang et al., 2024; Jain et al., 2024). From a task paradigm perspective, early attempts adopt open-set and domain generalization strategies (Acsintoae et al., 2022; Zhu et al., 2022; Jain et al., 2024). Then, Wu et al. (2024b) extends this paradigm with open-vocabulary VAD, enabling both detection and classification of unseen anomalies. However, these approaches implicitly assume a fixed anomaly definition and restrict model exposure to partial categories during training, unable to deal with the concept drift issue. Recent studies explore the dynamic anomaly definition: Cao et al. (2023); Cho et al. (2023); Aich et al. (2023) posit that anomaly is scene-dependent (e.g., identical behaviors classified differently across scenes), training models to infer scene-anomaly correlations from data, and Cho et al. (2024) trains dataset-specific classifiers. Despite these efforts, they lack the ability of user-customizable anomaly definition, limiting their applicability in open-world scenarios. Additionally, some recently developed MLLM-based methods (Tang et al., 2024; Yang et al., 2024a; Zanella et al., 2024; Zhang et al., 2025; Ye et al., 2025) have the potential to address open-world problems through prompt engineering, but they do not systemat-

ically study the issue of concept drift and generally require great computational costs. And we find that prompt engineering alone is unable to achieve satisfactory performance.

From a model design perspective, current advancements primarily adopt two ways: 1) data-driven approaches (Acsintoae et al., 2022; Zhu et al., 2022; Jain et al., 2024; Wu et al., 2024b) enhance generalization by utilizing more data, while 2) cross-modal alignment approaches (Chen et al., 2023; Wu et al., 2024c; Yang et al., 2024b) aim to construct more robust feature spaces by aligning vision and language. However, they neglect the problem of duration distribution shifts when leveraging more data and only align videos to class-level text embeddings without further fine-grained aligning.

Our work introduces a novel open-world VAD paradigm that allows users to flexibly define anomalies to guide detection, thereby avoiding concept drift. We implement this paradigm via a model featuring dynamic video synthesis and contrastive learning with hard negative mining. Inspired by data augmentation methods (Devries & Taylor, 2017; Ren et al., 2025) and contrastive learning methods (Radford et al., 2021), the dynamic video synthesis module synthesizes videos of variable durations to increase the diversity and coverage of temporal patterns, and the hard negative mining module increases the sample's quality to achieve fine-grained modal alignment.

## 3 PARADIGM: LANGUAGE-GUIDED OPEN-WORLD VIDEO ANOMALY DETECTION

We define open-world video anomaly detection as the task of identifying video frames containing abnormal patterns, where the definition of abnormality may change during testing. Abnormal patterns manifest as events, behaviors, or actions (e.g., running). In practice, the definition of anomalies may change as requirements change, influenced by cultural differences, policy updates, and specific environments. The user may expand the definition to detect new anomalies or narrow the definition to remove those of no interest, which causes the label of a particular pattern to change. For instance, while running is generally normal behavior, it becomes abnormal in libraries or offices. Based on these observations, we propose the definition-determined abnormality assumption:

**Assumption 1** (Definition-determined abnormality). *Let $V, Z, Y$ be random variables denoting the video, the definition, and the anomaly label, respectively. We assume that $Y$ is solely determined by $V$ and $Z$. That is, there exists a deterministic function $\mathcal{F}$ such that for all $v, z, y$,*

$$P(V = v, Z = z, Y = y) > 0 \implies y = \mathcal{F}(v, z).$$

**Proposition 1.** *Let $D$ be a random variable denoting the domain, where each value $d$ induces a joint distribution $P_d(V, Z, Y)$ (e.g., training domain or testing domain). For each domain $d$, let $P_d(\cdot)$ denote probabilities under the conditional distribution $P(\cdot \mid D = d)$. Under Assumption 1, for any two domains $d_1$ and $d_2$,*

$$P_{d_1}(Y \mid V, Z) = P_{d_2}(Y \mid V, Z).$$

**Proposition 2.** *Consider two domains, $d_1$ and $d_2$, where the anomaly definition shifts between domains, i.e., $P_{d_1}(Z \mid V) \neq P_{d_2}(Z \mid V)$. Suppose there exist at least one video $v^\star$ and one label $y^\star$ such that the conditional probability mass assigned to anomaly definitions that predict $v^\star$ as $y^\star$ differs across domains, i.e.,*

$$\sum_{z:\mathcal{F}(v^\star,z)=y^\star} P_{d_1}(Z = z \mid V = v^\star) \neq \sum_{z:\mathcal{F}(v^\star,z)=y^\star} P_{d_2}(Z = z \mid V = v^\star),$$

*then*

$$P_{d_1}(Y|V) \neq P_{d_2}(Y|V),$$

*which corresponds to the concept drift issue defined in Moreno-Torres et al. (2012)*

The proof of the above propositions is provided in Section I. We also discuss some special situation of the assumption in Section H.

Intuitively, we assume that the anomaly label of a video is determined solely by the video itself and the anomaly definition (Assumption 1). When the anomaly definition changes, the label of the same video is likely to change as well, which leads to concept drift (Proposition 2). In contrast, for any

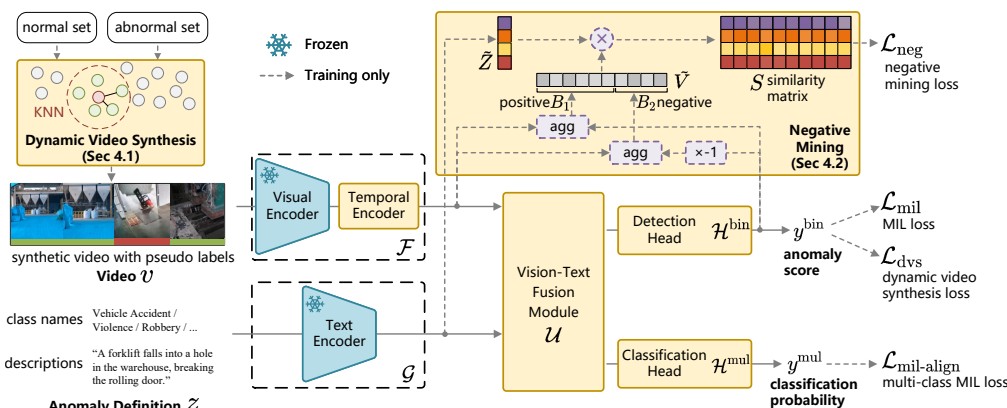

Figure 2: Architecture of our proposed LaGoVAD, which implement Eq. 2 by adding an anomaly definition branch ($z \rightarrow \mathcal{G} \rightarrow \mathcal{U}$). The model is trained with two novel regularization strategies: dynamic video synthesis $\mathcal{L}_{\text{dvs}}$ (4.1) and contrastive learning loss with negative mining $\mathcal{L}_{\text{neg}}$ (4.2).

fixed video and anomaly definition, the corresponding label remains unchanged regardless of how the underlying data distribution shifts, and thus eliminates concept drift (Proposition 1).

Existing methods can be seen as modeling $\Phi : V \rightarrow Y$ and performing detection based on a fixed definition $z$ sampled from $Z$, which faces the concept drift since $P(Y \mid V)$ may change:

$$\theta^\star = \arg\min_{\theta} \mathbb{E}_{(v,y)\sim P(V,Y)}[\mathcal{L}(\Phi(v;\theta,z),y)], \tag{1}$$

where $\theta$ denotes the parameters of the model $\Phi$, and $\mathcal{L}$ denotes the loss function. It is worth emphasizing that some methods that can detect unknown anomalies also belong to this paradigm, including open-set (Zhu et al., 2022; Acsintoae et al., 2022), domain generalization (Wang et al., 2024c) and open-vocabulary (Wu et al., 2024b) methods, because they assume a fixed category set under a specific definition and only a subset are available in training. Under their assumption, an abnormal pattern would never change to normal, and thus they are unable to deal with the concept drift.

In contrast, we propose a paradigm that directly models $\Phi : (V, Z) \rightarrow Y$, which could avoid the concept drift since $P(Y \mid V, Z)$ remains unchanged. It assumes a dynamic anomaly definition during training and conditions predictions on both the video and the definition. Formally,

$$\theta^\star = \arg\min_{\theta} \mathbb{E}_{(v,z,y)\sim P(V,Z,Y)}[\mathcal{L}(\Phi(v,z;\theta),y)]. \tag{2}$$

During training, the model $\Phi$ learns an optimal set of parameters $\theta$ that detect anomalies in video $v$ under the guidance of definition $z$. We later implement $z$ in the form of natural language, but theoretically, it could be images, videos, audio, or a learned embedding. It should be emphasized that although the new paradigm theoretically avoids concept drift, its practical effectiveness still depends on the model $\Phi$ and its parameters $\theta$.

## 4 METHOD: LAGOVAD

We implement the language-guided VAD paradigm (Eq. 2) via LaGoVAD. We first introduce the overall architecture, followed by details of two proposed regularization terms.

As illustrated in Fig. 2, we take video $v$ and anomaly definition $z$ as inputs. The video is synthesized by a non-parametric dynamic video synthesis module. The anomaly definition is a category set $z = \{z_0, z_1, \ldots, z_{C-1}\}$, where each class $z_i$ is defined by a class name or a description and $C$ is the number of categories in a certain definition. During training, we randomly choose either the class names or the anomaly descriptions within a batch as the definition. A video is considered as normal when the description does not belong to it. We extract and encode features of videos with $\mathcal{F}$, which includes a pretrained CLIP image encoder (Radford et al., 2021) and a Transformer-based temporal encoder. And the text features are extracted with CLIP text encoder $\mathcal{G}$. Then, the encoded features are fused by a Transformer-based fusion module $\mathcal{U}$. Finally, the fused features are fed into a binary

detection head $\mathcal{H}^{\text{bin}}$ to obtain the anomaly score $y^{\text{bin}} \in \mathbb{R}^{L \times 1}$ and a multi-classification head $\mathcal{H}^{\text{mul}}$ to obtain the classification probability $y^{\text{mul}} \in \mathbb{R}^{L \times C}$, where $L$ is the length of video. Formally,

$$[v, y^{\text{p}}] = \text{Synthesis}(N, A), \tag{3}$$

$$v^t = \mathcal{F}(v), \quad z^t = \mathcal{G}(z), \quad [v^u, z^u] = \mathcal{U}(v^t, z^t), \tag{4}$$

$$y^{\text{bin}} = \mathcal{H}^{\text{bin}}(v^u), \quad y^{\text{mul}} = \mathcal{H}^{\text{mul}}(v^u, z^u), \tag{5}$$

where $N, A$ are normal and abnormal video sets, $\text{Synthesis}(\cdot, \cdot)$ is the dynamic video synthesis module, $y^{\text{p}}$ is the pseudo-label generated during synthesis, $v^t, z^t$ are encoded features and $v^u, z^u$ are fused features.

During training, we optimize the model through four losses under weak supervision. Following (Wu et al., 2024b;c), we calculate multiple instance learning loss $\mathcal{L}_{\text{MIL}}$ (with $y^{\text{bin}}$) and MIL-align loss $\mathcal{L}_{\text{MIL-align}}$ (with $y^{\text{mul}}$) to optimize temporal binary detection and video-level multi-class classification. Our paradigm operates in multimodal joint spaces $P(V, Z, Y)$ that inherently suffer from exponentially decaying sample density, thereby inducing overfitting problems. Specifically, the algorithm may establish a wrong mapping or suppress a certain modality. Therefore, we leverage more diverse videos via a dynamic video synthesis loss $\mathcal{L}_{\text{dvs}}$ to learn better mappings. We also incorporate a contrastive learning loss with hard negative mining $\mathcal{L}_{\text{neg}}$ for better alignment. Formally,

$$\mathcal{L} = \mathcal{L}_{\text{MIL}} + \mathcal{L}_{\text{MIL-align}} + \mathcal{L}_{\text{dvs}} + \mathcal{L}_{\text{neg}}. \tag{6}$$

This work prioritizes addressing the challenge of concept drift over designing complex architectures. Consequently, we adopt a simple but effective network. The proposed two regularizers are independent in design, which could be seamlessly integrated into more sophisticated architectures.

## 4.1 DYNAMIC VIDEO SYNTHESIS

In real-world scenarios, anomalies typically occupy only a small portion of a lengthy video, whereas current datasets predominantly contain videos with high anomaly ratios due to web-sourced data limitations. To mitigate this bias, we dynamically synthesize videos with varying durations and compute a loss based on the pseudo label generated during synthesis. The module initially determines whether to generate a normal or abnormal video, followed by specifying the number of segments. In particular, when the number is 1, it indicates that no synthesis is performed. It then selects an anchor video and randomly selects similar videos from k-nearest neighbors to construct a semantically coherent sequence, where the anchor's position is transformed to a binary pseudo label $y^{\text{p}} \in \{0, 1\}^L$, where $L$ denotes the feature length. The visual representation used for semantic retrieval remains unchanged from that of the backbone, since segments retrieved with high similarity are largely indistinguishable from the model's perspective. Notably, the distance metrics required for retrieval are pre-computed, effectively reducing computational overhead during training. Finally, a dynamic video synthesis is calculated as:

$$\mathcal{L}_{\text{dvs}} = -\hat{y} \log \sum_{i \in \Omega_k^a} \sigma(y_i^{\text{bin}})/k - (1 - \hat{y}) \log(1 - \sum_{i \in \Omega_k^n} \sigma(y_i^{\text{bin}})/k) \tag{7}$$

$$- \sum_i^L y_i^{\text{p}} \log \sigma(y_i^{\text{bin}})/L, \tag{8}$$

where $\sigma$ denotes the Sigmoid function, $\hat{y}$ denotes the video-level ground truth, $\Omega_k^a$ and $\Omega_k^n$ are indices of Top-K scores of synthetic abnormal and normal videos, respectively.

## 4.2 CONTRASTIVE LOSS WITH HARD NEGATIVE MINING

Given the ambiguous boundary between normal and abnormal frames in anomaly videos, we incorporate contrastive learning with hard negative mining as a regularization term to enhance their discriminability. Specifically, we first aggregate the frame-level visual features into video-level features with binary abnormal scores as weights:

$$\tilde{v}^{\text{pos}} = \sum_i^L v_i^t \frac{\exp\left(y_i^{\text{bin}}/\eta\right)}{\sum_j^L \exp\left(y_j^{\text{bin}}/\eta\right)}, \quad \tilde{v}^{\text{neg}} = \sum_i^L v_i^t \frac{\exp\left(-y_i^{\text{bin}}/\eta\right)}{\sum_j^L \exp\left(-y_j^{\text{bin}}/\eta\right)}, \tag{9}$$

where $v_i^t$ denotes the $i$-th feature in $v^t$, $\eta$ denotes the temperature, $\tilde{v}^{\text{pos}}, \tilde{v}^{\text{neg}}$ denote the aggregated foreground/background feature. The background feature in an abnormal video is the normal part of

it, which could be considered as the hard negative to its corresponding anomaly description. And the selection of hard negatives can be adjusted through the temperature coefficient $\eta$. Then, we obtain $\tilde{v}^{\mathrm{pos}}$ of all samples and $\tilde{v}^{\mathrm{neg}}$ of only abnormal samples in a batch, forming $\tilde{V} \in \mathbb{R}^{(B_1+B_2) \times E}$, where $B_1$ is the batch size, $B_2$ is the number of abnormal videos in a batch, and $E$ is the feature dimension. We also obtain the text features before fusing, forming $\tilde{Z} \in \mathbb{R}^{B_2 \times E}$. The contrastive loss is as follows:

$$\mathcal{L}_{t \to v} = -\sum_{i}^{B_2} \log \frac{\exp\left(S_{i,i}/\tau\right)}{\sum_{j}^{B_1+B_2} \exp\left(S_{j,i}/\tau\right)}, \quad \mathcal{L}_{v \to t} = -\sum_{j}^{B_2} \log \frac{\exp\left(S_{j,j}/\tau\right)}{\sum_{i}^{B_2} \exp\left(S_{i,j}/\tau\right)}, \qquad (10)$$

$$\mathcal{L}_{\mathrm{neg}} = \mathcal{L}_{t \to v} + \mathcal{L}_{v \to t}, \qquad (11)$$

where $S = Norm(\tilde{V}) \times Norm(\tilde{Z})$, $Norm$ is L2 normalization and $\tau$ denotes the temperature.

During inference, the user can input either descriptions or class names as the anomaly definition. For the classification head, we select the minimum value of the normal class and the maximum value of the abnormal class over the temporal axis and use these values after applying Softmax as probabilities. More architecture details are provided in *supp* (Sec. C.1).

## 5 DATASET: PREVAD

### 5.1 DATA CURATION PIPELINE

We propose PreVAD—a large-scale pretraining VAD dataset to provide diverse $(v, z, y)$ triples for training, which is collected through a scalable curation pipeline. The proposed pipeline encompasses three stages: source, cleansing, and annotation.

We aggregate videos from three sources: First, we retrieve anomaly videos from existing large-scale video-text datasets (Xu et al., 2016; Wang et al., 2019; Liu et al., 2025; Zhu et al., 2024a) utilizing their text annotation. Second, we expand the collection through curated web resources, including 1) accident compilations and fail videos; 2) driving and travel vlogs; 3) violence recognition videos. Last, we obtain normal surveillance videos from YouTube streams and traffic camera streams.

In the cleansing stage, we first remove irrelevant segments such as intros and outros with automatic tools. Next, a multimodal LLM (MLLM) generates detailed video descriptions, and a vision-language model verifies the consistency between the descriptions and video. Finally, an LLM evaluates the descriptions to confirm the presence of anomalies, decreasing hallucinations.

In the annotation stage, we employ a hybrid human-AI approach. First, we annotate each video with a category label. Then, using this label as a constraint, we prompt an MLLM to generate fine-grained descriptions of the anomalies, ensuring focused and relevant output. We also conduct frame-level annotations for the validation set. Notably, we do not additionally label a test set, as we will conduct zero-shot evaluations on other existing VAD datasets. More details can be found in the *supp* Sec. D.

### 5.2 DATASET STATISTICS

Our dataset stands out for its large scale, wide variety of anomalies, and high-quality descriptions.

**Scale.** PreVAD comprises 35,279 videos, spanning a total duration of 209.5 hours, with 11,979 abnormal videos and 23,300 normal videos, partitioned as shown in Fig. 3a, which is the largest video anomaly dataset up to now, as compared in Fig. 3b.

**Anomaly Types.** Our dataset's diversity stems from a hierarchical taxonomy with 7 first-level categories (i.e., *Violence, Vehicle Accident, Fire-related Accident, Robbery, Daily Accident, Animal-related Violence, Production Accident*) and 35 subcategories (e.g., *carjacking, mugging, sport fail, war*), spanning from minor (e.g., fall to the ground) to severe (e.g., shooting) anomalies, which covers a broad range of scenarios.

**Anomaly Descriptions.** Each abnormal video is annotated with a text description, which has a total vocabulary size of 5,298 words and an average of 22.9 words per description. As shown in Fig. 3c, our annotation accurately describes the abnormal objects and behaviors in a fine-grained manner.

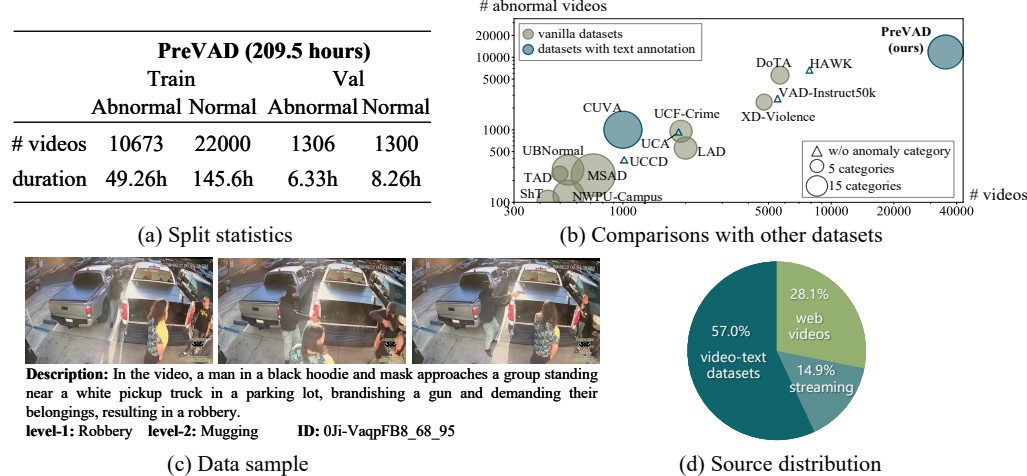

Figure 3: The statistics, comparisons and a data sample of the proposed PreVAD dataset.

**Sources.** As shown in Fig. 3d, most of the videos are from existing video-text datasets or streaming, significantly reducing the overhead of manual clipping and retrieval. PreVAD also obtains videos independently without merging existing VAD datasets, enabling cross-dataset validation as a new generalization benchmark. We provide more details of PreVAD in the *supp* Sec. D.

## 6 EXPERIMENTS

### 6.1 EXPERIMENT SETUP

**Datasets** We conduct comprehensive zero-shot evaluations across seven datasets: UCF-Crime (UCF) (Sultani et al., 2018), XD-Violence (XD) (Wu et al., 2020), MSAD (Zhu et al., 2024b), UB-Normal (UBN) (Acsintoae et al., 2022), DoTA (Yao et al., 2022), TAD (Lv et al., 2021), and LAD (Wan et al., 2021), which encompass diverse anomaly types. The validation set of our proposed PreVAD is also utilized for in-domain analysis and ablations. More details are in *supp* Sec. E.1.

**Evaluation Protocols** We evaluate the open-world capability with two zero-shot protocals: **Protocol 1**: Testing on multiple test sets separately, each representing a distinct scenario (e.g., TAD for traffic scenarios), which evaluates the overall performance under concept drift, unseen categories, feature distribution shifts, etc. **Protocol 2**: Testing on a dataset with varying anomaly definitions, where in each definition only a subset of anomaly categories is considered as abnormal. Such variations between subsets simulate variable user requirements in real-world applications. The final performance is averaged across five such definitions (denoted as drift@5), specifically evaluating the model's robustness to concept drift. The differences of test sets and the selected subsets are detailed in *supp* Sec. E.2. During evaluation, we use manual designed prompts based on the class name of the corresponding dataset as the anomaly definition.

**Comparative Methods** For Protocol 1, we compare against traditional methods (PEL (Pu et al., 2024), VadCLIP (Wu et al., 2024c)), along with scene-dependent (CMRL (Cho et al., 2023)), zero-shot (LAVAD (Zanella et al., 2024)), open-vocabulary (OVVAD (Wu et al., 2024b)), and multi-domain generalization (MultiDomain (Cho et al., 2024)) approaches. We also include open-vocabulary action recognition methods (ActionCLIP (Wang et al., 2021), ViFi-CLIP Rasheed et al. (2023)) for multi-class comparison. For Protocol 2, as most methods do not support user-provided anomaly definition, comparisons are primarily made with LLM-based (Qwen2-VL (Wang et al., 2024a), Qwen2.5-VL (Bai et al., 2025), LAVAD, HolmesVAU Zhang et al. (2025)) and multi-modal methods (VadCLIP). All results are based on their open-source codes and weights, detailed in Sec. F.

**Metrics** For binary detection metrics and without additional annotations, we follow existing works using Average Precision (AP) for XD-Violence and using Area Under the Curve of the frame-level

Table 2: Comparison in temporal binary anomaly detection under Protocol 1. Results marked with † are taken from their publications and results marked with ‡ are from Zanella et al. (2024).

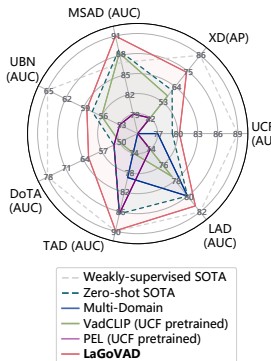

| Methods | Training-set | UCF (AUC) | XD (AP) | MSAD (AUC) | UBN (AUC) | DoTA (AUC) | TAD (AUC) | LAD (AUC) |
|---|---|---|---|---|---|---|---|---|
| | | | | | Test-set | | | |
| OVVAD† | AIGC+XD | 82.42 | - | - | - | - | - | - |
| **LaGoVAD** | **PreVAD**+XD | 82.81 | - | - | - | - | - | - |
| OVVAD† | AIGC+UCF | - | 63.74 | - | - | - | - | - |
| **LaGoVAD** | **PreVAD**+UCF | - | 76.28 | - | - | - | - | - |
| CLIP‡ | - | 53.16 | 17.83 | - | - | - | - | - |
| LLaVA1.5‡ | - | 72.84 | 50.26 | - | - | - | - | - |
| LAVAD† | - | 80.28 | 62.01 | - | - | - | - | - |
| CMRL† | UCF | - | 46.74 | - | - | - | - | - |
| MultiDomain† | Multiple | 78.55 | - | - | - | - | 79.21 | 77.36 |
| PEL | UCF | - | 43.53 | 79.82 | 54.02 | 53.05 | 86.27 | 69.99 |
| PEL | XD | 54.52 | - | 68.25 | 49.55 | 44.97 | 43.02 | 30.82 |
| VadCLIP | UCF | - | 58.29 | 88.09 | 56.24 | 50.93 | 74.46 | 74.29 |
| VadCLIP | XD | 80.16 | - | 88.48 | 57.41 | 49.00 | 83.56 | 74.46 |
| VadCLIP | **PreVAD** | 79.37 | 67.43 | 89.79 | 55.66 | 50.59 | 85.96 | 75.02 |
| **LaGoVAD** | **PreVAD** | **81.12** | **74.25** | **90.41** | **58.07** | **62.60** | **89.56** | **78.91** |

Table 3: Comparison in video-level multi-class classification on UCF-Crime and XD-Violence under Protocol 1. All the methods employ the same CLIP variant.

| Method | Training | UCF Acc. | UCF F1 | XD Acc. | XD F1 |
|---|---|---|---|---|---|
| CLIP | - | 19.31 | 12.08 | 56.25 | 45.04 |
| ActionCLIP | K400 | 18.62 | 16.12 | 38.75 | 37.11 |
| ViFi-CLIP | K400 | 20.34 | 15.67 | 53.75 | 50.33 |
| VadCLIP | UCF | - | - | 46.38 | 26.16 |
| VadCLIP | XD | 38.28 | 10.52 | - | - |
| VadCLIP | **PreVAD** | 45.52 | **17.81** | 71.38 | 57.99 |
| **LaGoVAD** | **PreVAD** | **51.72** | 16.64 | **78.13** | **63.80** |

Table 4: Comparison in temporal binary anomaly detection on XD and MSAD under Protocol 2, specifically evaluating robustenss to concept drift. The model marked with † is trained on PreVAD.

| Method | XD-drift@5 AUC | XD-drift@5 AP | MSAD-drift@5 AUC | MSAD-drift@5 AP |
|---|---|---|---|---|
| Qwen2-VL-7B | 60.4 | 17.5 | 65.4 | 22.9 |
| Qwen2.5-VL-7B | 62.7 | 20.6 | 63.1 | 22.4 |
| VadCLIP † | 81.3 | 35.8 | 85.2 | 18.8 |
| HolmesVAU | - | - | 84.3 | 34.3 |
| LAVAD | 81.7 | 34.8 | 72.2 | 31.5 |
| **LaGoVAD** | **85.7** | **37.1** | **85.6** | **40.1** |

receiver operating characteristic (AUC) for other datasets. For multi-class classification metrics, we use multi-class accuracy and F1-score on both abnormal and normal videos.

Details of implementation of our method are in *supp* Sec. C.2.

## 6.2 COMPARISON WITH STATE-OF-THE-ARTS

Under the comprehensive evaluation of Protocol 1 (Tabs. 2,3), our approach surpasses others across all datasets, which includes comparisons with related methods in open-vocabulary setting (OVVAD, ActionCLIP, ViFi-CLIP), cross-domain setting (MultiDomain), or scene-dependent setting (CMRL). Notably, on XD-Violence, it achieves improvements of 20% and 32% in detection and classification, respectively. Under the concept drift evaluation of Protocol 2 (Tab. 4), LaGoVAD achieves better performance than multi-modal methods and LLM-based methods, while avoiding the significant computational overhead from huge parameters.

## 6.3 ABLATION STUDIES

**Module Effectiveness** We report ablation studies in Tab. 5. Removing either $\mathcal{L}_{dvs}$ or $\mathcal{L}_{neg}$ led to a noticeable degradation in detection and classification performance. When both are removed, the model exhibits a significant decline in zero-shot performance. When disabling the language guidance, we followed approaches in (Wu et al., 2024c;b) to place the fusion module after the detection stage, which does not condition detection results on the given text. Ex-

Table 5: Ablation on each component. Lang-guided: language guiding. Det. Avg.: average zero-shot temporal detection performance on seven datasets. Cls. Avg.: average zero-shot multi-classification performance on UCF and XD.

| Lang-guided | $\mathcal{L}_{dvs}$ | $\mathcal{L}_{neg}$ | PreVAD | Det. Avg. | Cls. Avg. |
|---|---|---|---|---|---|
| ✓ | ✓ | ✓ | **69.98** | **76.42** | **52.57** |
| ✓ | | ✓ | 65.73 | 73.51 | 51.73 |
| ✓ | ✓ | | 68.92 | 73.96 | 51.85 |
| ✓ | | | 67.35 | 71.31 | 48.81 |
| | ✓ | ✓ | 69.87 | 73.84 | 46.23 |

periment shows that without language guidance, cross-domain performance decreased significantly, which indicates that the conventional paradigm lack the capacity to incorporate user-defined guidance for detection, thereby limiting their adaptability to open-world scenarios. More ablations of modules and data are in the *supp* Sec. G.

**Dataset & Architecture Effectiveness**  To quantify dataset and architecture impacts, we compare VadCLIP trained on PreVAD. The results reveal that the model trained on PreVAD outperforms the one trained on UCF-Crime by 14% in detection (average metric on six other datasets) and 88% in classification (average metric on XD-Violence) while also surpassing the one trained on XD-Violence by 7.6% in detection and 44% in classification, respectively. This substantial margin validates that a larger and more diverse dataset can significantly improve zero-shot performance. When trained on PreVAD, our LaGoVAD achieves consistent improvements over VadCLIP, with gains of 7.2% in average detection performance across seven datasets and 2.8% in classification across on two datasets. This confirms the superiority of our approach in open-world scenarios.

## 6.4 Qualitative Results

Fig. 4 visualizes the performance under concept drift. The conventional method (VadCLIP) fails to handle dynamic definitions, producing same scores under the training definition. Although LLM based methods (LAVAD, Qwen2.5-VL) can take definition prompts, LAVAD fails to recognize the anomaly due to its limited understanding of dynamic events. Qwen2.5-VL recognizes it but cannot localize it precisely. Our method, in contrast, adapts to the dynamic definition and achieves precise anomaly localization.

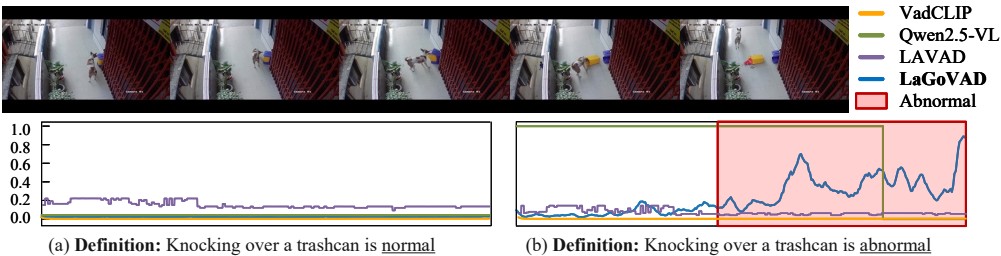

(a) **Definition:** Knocking over a trashcan is normal  (b) **Definition:** Knocking over a trashcan is abnormal

Figure 4: Visualization of different methods under concept drift. *Knocking over a trashcan* is considered normal in (a) but abnormal in (b). All models are prompted with the corresponding definition.

## 7 Conclusion

In this work, we propose a novel paradigm, language-guided open-world video anomaly detection, to deal with concept drift in the open-world scenario. It assumes that the definition of anomaly is dynamic and models it as a stochastic variable input to the network. To support training this model, we build a large-scale video anomaly dataset that is annotated by multi-level taxonomy and anomaly descriptions. We empirically verify the effectiveness of the proposed framework through state-of-the-art zero-shot performance and sufficient ablations on seven datasets.

## Acknowledgement

This work is supported by the Fundamental Research Funds for the Central Universities (No.CUC25GT29,CUC25SG012,CUC25SG008,CUC25QT17).

## Reproducibility Statement

To facilitate the reproducibility of our work, we have fully prepared our codes and pre-trained weights to release. The repository includes detailed instructions for installing dependencies, training models from scratch, and evaluating on the benchmark datasets. We provide all hyperparameters used in our experiments to ensure consistent results. The experiments were conducted on multi-

ple kinds of NVIDIA GPUs with consistent results. The code and dataset are now available at `https://github.com/Kamino666/LaGoVAD-PreVAD`.

ETHICS STATEMENT

We have read and adhere to the ICLR Code of Ethics. Our work introduces a novel method for video anomaly detection in open-world settings, with potential applications in intelligent surveillance and traffic safety systems. While this technology offers beneficial uses, we acknowledge that it could also be misused—for instance, in enabling large-scale surveillance when integrated with person identification technologies. Following common practices in the field, the video data used in this study are sourced from publicly available platforms and we only provide annotations, features, source URLs and download scripts. We acknowledge potential legal risks associated with the use of the data, as researchers using data from certain sources may be undertaking legal risk if violating terms of the license. The detailed dataset statement is provided in Sec. D.1. We declare no potential conflicts of interest.

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

## A    DECLARATION OF LLM USAGE

LLM is only used for writing, editing, and formatting, which does not impact the core methodology, scientific rigorousness, or originality of this research.

## B    LIMITATION AND FUTURE WORK

While our work introduces a novel paradigm for addressing concept drift in open-world video anomaly detection, we acknowledge that limitation exists. The current architecture instantiates the paradigm through a simplified design, leaving room for architectural refinements to better capture temporal dependencies and multimodal interactions. For future work, researchers may establish comprehensive benchmarks beyond the current evaluation to systematically evaluate prompt compliance and open-world capability. Moreover, through our proposed pipeline, a larger dataset could be collected to further boost the performance.

## C    DETAILS OF LaGoVAD

### C.1    ARCHITECTURE

**Dynamic Video Synthesis**    In practical scenarios, abnormal events typically occupy a small proportion of the total video duration. However, since people often edit videos to highlight events of interest before uploading them to the Internet, web-sourced videos generally exhibit high anomaly ratios. For instance, the test sets of MSAD, PreVAD, and LAD contain 42%, 39%, and 38% of videos, respectively, where abnormal events occupy over 70% of the total duration. This data characteristic leads to the loss of normal contextual information, consequently hindering the model's ability to learn normal-anomaly boundaries. Our proposed dynamic video synthesis addresses this issue by concatenating semantically similar video segments to reconstruct normal contextual information, with the detailed workflow described as follows.

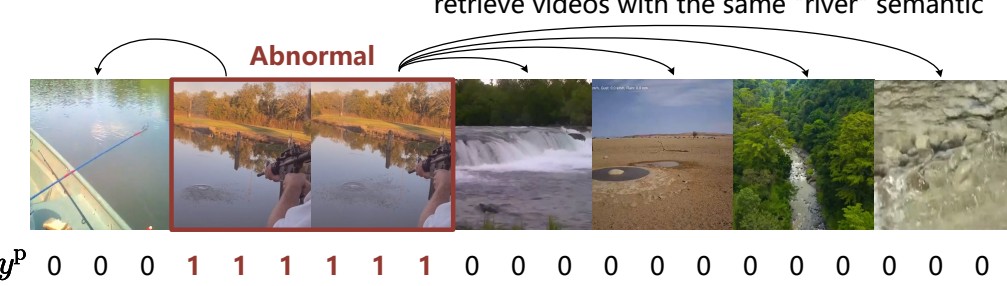

Figure A: An example of a synthesized video with our proposed dynamic video synthesis.

Dynamic video synthesis comprises a synthesis module and its corresponding pseudo-label loss. Algorithm 1 illustrates the workflow of our synthesis module. The process begins by randomly selecting either a normal or abnormal video as the anchor $\tilde{v}$ (determining whether the synthesis target is normal or abnormal). Subsequently, it randomly determines the number of synthetic segments $m$ and their insertion positions $j$. The remaining positions are then populated by randomly selecting videos from the $N$ nearest neighbors in the normal video set. For nearest neighbor computation, we employ `CLIP-ViT-B/16` features extracted from the central frame of each video, using cosine similarity as the distance metric. Unlike the 10-crop approach that pre-generates the augmented data, our method dynamically synthesizes samples during training. To accelerate training efficiency, we pre-compute the $N$ nearest neighbors for each sample in advance.

Since the positions of abnormal segments are known during synthesis, we sequentially construct pseudo-labels $y^{\mathrm{p}} \in \mathbb{R}^L$, where $L$ denotes the total length of the synthesized video features. For synthesized normal videos, all elements in $y^{\mathrm{p}}$ are set to 0. For synthesized abnormal videos, $y^{\mathrm{p}}$

---

**Algorithm 1** Dynamic Video Synthesis Module

---

**Input:** Normal video set $N$, Abnormal video set $A$, Synthesis probability $\theta \in [0, 1]$, Normal video probability $\alpha \in [0, 1]$, Maximum number of segments $\delta_m \in \mathbb{N}^+$, Number of nearest neighbors $n \in \mathbb{N}^+$

**Output:** Synthesized video segment sequence $v$

1: **Initialize:** video sequence $v$
2: Sample $p_1, p_2 \sim U(0, 1)$
3: **if** $p_1 > \theta$ **then**
4:     $m \leftarrow 1$
5: **else**
6:     $m \leftarrow \text{Randint}(1, \delta_m)$
7: **end if**
8: **if** $p_2 > \alpha$ **then**
9:     Sample $\tilde{v}$ from $A$
10: **else**
11:     Sample $\tilde{v}$ from $N$
12: **end if**
13: $j \leftarrow \text{Randint}(1, m)$
14: **for** $i = 1$ to $m$ **do**
15:     **if** $i = j$ **then**
16:         Append $\tilde{v}$ to $v$
17:     **else**
18:         Sample $v_n$ from $\text{NearestNeighbor}(N, \tilde{v}, n)$
19:         Append $v_n$ to $v$
20:     **end if**
21: **end for**
22: **return** $v$

---

takes the value 1 in abnormal intervals and 0 elsewhere. Fig. A shows an example. Using these pseudo-labels, we compute the following loss function:

$$\mathcal{L}_{\text{dvs}} = - \hat{y} \log \sum_{i \in \Omega_k^a} \sigma(y_i^{\text{bin}})/k \tag{12}$$

$$- (1 - \hat{y}) \log(1 - \sum_{i \in \Omega_k^n} \sigma(y_i^{\text{bin}})/k) \tag{13}$$

$$- \sum_i^L y_i^{\text{p}} \log \sigma(y_i^{\text{bin}})/L, \tag{14}$$

where $\sigma$ denotes the Sigmoid function, $\hat{y}$ denotes the video-level ground truth, $L$ denotes the feature length, $\Omega_k^a$ and $\Omega_k^n$ are indices of Top-K scores of synthetic abnormal and normal videos, respectively. The first two terms can be viewed as variants of MIL loss that constrain the selection of abnormal instances to regions with pseudo-label value 1, while the last term directly leverages pseudo-labels for supervised learning.

**Temporal Encoder**   We employ a vanilla Transformer with rotary positional encoding (Su et al., 2024) as our temporal encoder.

**Fusion**   As illustrated in Fig. B, our vision-text modality fusion adopts a simple co-attention Transformer architecture. Each modality branch contains a cross-attention layer (CA) followed by a feedforward network (FFN), where the current modality serves as the query in the cross-attention while the other modality provides keys and values.

**Heads**   As illustrated in Fig. B, the temporal temporal detection head utilizes a 1D convolutional network with replicate padding, which processes both pre-fusion and post-fusion features through separate pathways to generate language-guided and language-agnostic anomaly scores. These scores are subsequently fused via a learnable parameter to produce final predictions. The video-level multiclass classification head computes a similarity matrix between linearly projected representations of both modalities.

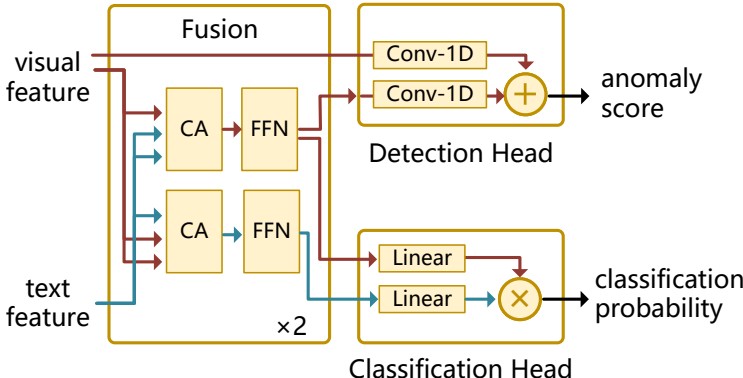

Figure B: Detail architecture of fusion module and heads.

## C.2 IMPLEMENTATION

All experiments could be conducted on a single NVIDIA RTX4090 GPU. We use PyTorch and PyTorch-Lightning as the code architecture. We set the temperature factors $\tau, \eta$ as 0.02 and employ a base hidden size of 512. The temporal encoder and the fusion module both have 2 layers, and the detection head uses a single layer with a kernel size of 9. For dynamic video synthesis, we set $\theta = 0.7, \alpha = 0.5, \delta_m = 5, n = 200$. We use AdamW as the optimizer with a batch size of 64 and a learning rate of 0.00005. The model is trained for 40 epochs. We do not use tricks like 5-crop, 10-crop, or score smooth, etc.. We take a sample every 8 frames, except for the DoTA dataset, which provides 10fps extracted frames.

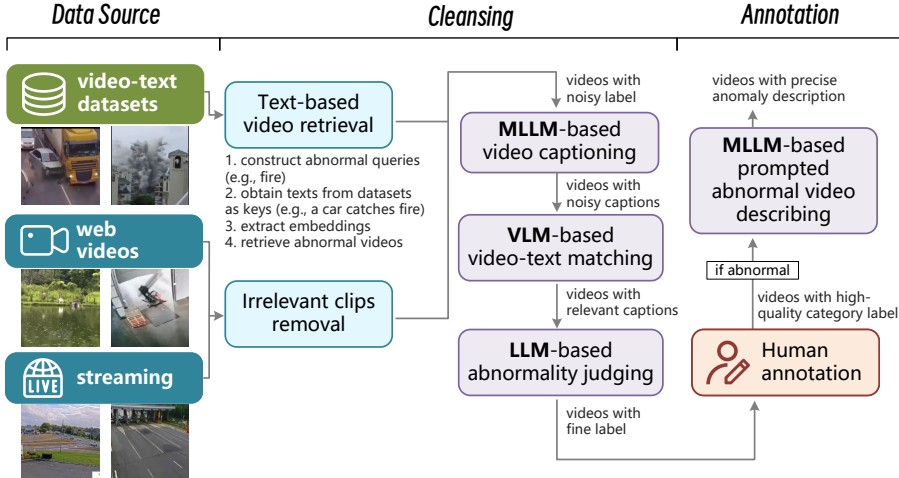

Figure C: The data curation pipeline for our dataset. It includes three phases: data source, cleansing, and annotation.

## D DETAILS OF PREVAD

### D.1 DATASET STATEMENT

The dataset used in this paper is a curated collection constructed by the authors to support research on video anomaly detection. The released resources consist of the authors' annotations and supporting files associated with videos collected from publicly accessible sources.

Table A: Details about the data sources used by PreVAD.

| Source | | Notes |
|---|---|---|
| video-text datasets | VIDAL-10M (Zhu et al., 2024a) | The VIDAL-10M dataset comprises 3 million pairs of video-language data crawled from YouTube Shorts and Freesound. It includes abnormal videos of accidents and normal videos of everyday events taken by people with their smart phones. |
| | MSR-VTT (Xu et al., 2016) | The MSR-VTT dataset is a video captioning dataset that comprises 10K videos with 20 captions per video. The videos are crawled from YouTube and are annotated by crowdsourcing. It contains many domains, including human activities, tutorials, games, etc. We mainly use them as normal data. |
| | VATEX (Wang et al., 2019) | The VATEX dataset is also a video captioning dataset with 35K videos, each having 10 manually labeled captions. Its videos are from the Kinetics dataset, and thus are mostly diverse human activities. We mainly use them as normal data. |
| | VALOR (Liu et al., 2025) | The VALOR dataset is a video captioning dataset that focuses on audio-based description. The authors release a subset of 32K videos, each with one manually annotated caption. The videos are from AudioSet, which covers human activities and natural scenes. We mainly use them as normal data. |
| web videos | collections | There are dedicated channels on YouTube and Bilibili that compile a variety of accident videos for the purposes of entertainment or education (e.g., FailsArmy, ASP). We clip segments from them as abnormal data. |
| | long videos | Some channels post hour-long videos, the content of which includes first-person-perspective driving, first-person-perspective traveling, and factory production records, etc. We randomly clip segments from such videos as normal data. (e.g., Driving) |
| | RWF-2000 (Cheng et al., 2020) | The RWF-2000 dataset is a violence recognition dataset that includes violent behaviors from a surveillance perspective, and we incorporate it into our dataset. |
| streaming | YouTube streams | On YouTube, there are numerous 24-hour surveillance live streams aimed at promoting tourism. We searched for such content using the keyword webcam and recorded segments at different times as normal data from a surveillance perspective. (e.g., Manchester UK Webcams) |
| | traffic cameras | With the development of intelligent transportation, many countries have deployed traffic cameras on highways and made their real-time footage publicly available on the Internet to help the public better plan their travel. We also recorded these as normal data. (e.g.,California Department of Transportation, Shanghai Municipal Transportation Commission) |

PreVAD is released under the Creative Commons Attribution-NonCommercial 4.0 International (CC BY-NC 4.0) license[1]. This license applies only to the materials created and released by us, including annotations, metadata, and related supporting files, and does not apply to the original video content, which remains subject to the license terms and terms of service of the corresponding source platforms or original datasets.

**Copyright and Terms of Service.** We respect the copyright policies and terms of service of the hosting platforms and source datasets. Accordingly, we do not redistribute raw video files or annotations originating from other datasets. Instead, PreVAD provides only: 1) video URLs for online resources; 2) video IDs for videos originating from other datasets; 3) our annotations; 4) download scripts; and 5) model-extracted features Radford et al. (2021). The referenced videos were publicly accessible at the time of collection and were not collected from private or access-restricted sources. We do not claim any ownership over the original video content and do not redistribute it.

**Privacy.** We do not provide video metadata such as uploader identities or account IDs. In addition, during feature extraction, we apply blurring or other anonymization techniques to personally identifiable information (PII) when present, so that the released features do not retain such sensitive information.

**Legal Risk Acknowledgement.** If any creator or rights holder requests removal, we will promptly remove the corresponding links and annotations from the released dataset. We acknowledge that some source videos may be subject to platform-specific terms, dataset-specific licenses, or other legal restrictions. Researchers who download or use such videos are responsible for ensuring that their use complies with the applicable terms and laws, and they bear any associated legal risks.

The annotators participated voluntarily and were compensated at a rate substantially above the locally mandated minimum wage.

---

[1]https://creativecommons.org/licenses/by-nc/4.0/

### D.2 PIPELINE

Fig. C illustrates the proposed data curation pipeline, leveraging publicly accessible sources for scalability and reproducibility, integrating multimodal foundation models (LLMs, MLLMs, VLMs) for intelligent cleaning, and employing a hybrid human-AI annotation framework to ensure quality while reducing costs.

**Source** Tab. A lists the data source details for PreVAD. An important issue faced by video anomaly detection for many years is data scarcity, and we exploit multiple data sources. To the best of our knowledge, we are the first to enumerate and analyze the data sources in detail. We mainly collected the data in 2024. All data sources are publicly available. Using these datasets, future works may construct a larger scale dataset.

**Cleansing** During the text-based video retrieval, we employ `all-MiniLM-L6-v2` and `CLIP-Large Text Encoder` to extract text embeddings of a set of abnormal queries (e.g., a fire broke out in the building) and captions in existing video-text datasets as keys. Then, we use FAISS (Douze et al., 2024) to retrieve abnormal captions and obtain their corresponding videos. We preliminarily retrieved over 55K videos in this phase, among which only nearly 75% of videos are available when downloading. During the irrelevant clips removal, we employ PySceneDetect[2] to split segments of the video and then automatically filter them with predefined rules of duration, RGB histogram, etc.. During the MLLM-based video captioning phase, we prompt `Qwen2-VL-72B-Instruct` with:

> *You are a helpful video describing assistant. You are good at English communication. Please describe the given video in detail. Just give the description without any other output.*

to obtain captions of existing videos for later filtering. During the VLM-based video-text matching, we utilize `InternVideo2` (Wang et al., 2024b) and `VALOR` (Liu et al., 2025) to calculate the matching scores of video-text pairs to remove irrelevant pairs with a threshold. During the LLM-based abnormality judging, we prompt `Qwen/Qwen2-72B-Instruct` with:

> *Below is the JSON format metadata of the title and content information of a video. I need you to determine if there are any anomalies in the video. Anomalies include:*
> *- any type of accident*
> *- ...*
> *Note, you need to output two lines, the first line is a brief one-sentence analysis, and the second line is a judgment of 'abnormal' or 'normal'. If you cannot provide analysis due to ethical guidelines, please just output 'abnormal'.*

to filter videos with wrong abnormality labels. After cleansing, we could obtain videos and corresponding video-level binary labels.

**Annotation** During the human annotation phase, we asked 7 annotators with expert knowledge for video anomaly detection to label each abnormal video with one video-level category among our taxonomy. They are also responsible for removing videos of low quality (e.g., anomalies far from the real world, excessive video effects or overlays of text, excessively short videos, and videos containing clips of multiple different scenes). For the validation set, we let two annotators independently annotate the segments of the selected abnormal videos and take the average as the final result. If the annotation difference between the two people is too large, a team leader will recheck. During the MLLM-based prompted abnormal video describing phase, we prompt `Qwen2-VL-72B-Instruct` with:

> *You are a helpful video describing assistant. You are good at English communication. Please describe the '{VIDEO LABEL}' event in the given video in detail in one sentence. The description should focus on the incident. Just describe without any other output.*

---

[2]github.com/Breakthrough/PySceneDetect

to obtain precise anomaly description. With this constrained prompt, the descriptions would be more related to the anomaly, therefore, are suitable for being anomaly definitions.

To quantify the quality of the annotations, we conduct a subjective quality assessment. We randomly sampled 200 videos, and asked three annotators to rate on a 1–5 scale independently, with the scoring criteria showning in Tab. B. On average, the descriptions generated without constrained prompt receive a score of 3.5, while those generated with constrained prompt receive a score of 4.3, demonstrating the effectiveness of the constrained prompt. We also assess the description annotations of HIVAU-70K dataset (Zhang et al., 2025) with the same criteria, which receive a score of 4.4, indicating the annotation noise is consistent with that of other works. We also calculate the intraclass correlation coefficient (ICC) (Shrout & Fleiss, 1979) to quantify inter-annotator agreement. The ICC(2,k) value was 0.799, indicating a high level of consistency across annotators.

Table B: Scoring criteria for evaluating the consistency between description and video.

| Score | Criteria |
| --- | --- |
| 5 | The text fully matches the video content, providing an accurate, complete, and natural description of the abnormal event. |
| 4 | The text is generally consistent with the video, with only minor deviations that do not affect overall understanding. |
| 3 | The text captures the main content but contains notable omissions or partial inaccuracies. |
| 2 | The text largely fails to match the video, capturing only limited relevant information or showing clear misinterpretations. |
| 1 | The text does not match the video content at all and fails to reflect the actual video. |

### D.3 MORE STATISTICS

Fig. Da provides the taxonomy of PreVAD, with Fig. Db and Fig. Dc show the distribution of categories. Fig. Dd provides a detailed source distribution visualization. Fig. De shows the proportion of anomaly in annotated abnormal videos and Fig. Df presents the distribution of video duration in seconds.

### D.4 EXAMPLES

We show examples of videos for each category and annotated examples in Figs. G,H. Our data contains a variety of high-quality anomaly videos, and the human-AI annotation pipeline can obtain accurate anomaly descriptions at a low cost.

## E DETAILS OF EVALUATION

### E.1 TEST SETS

Tab. C summarizes the number of test videos, anomaly categories, and other relevant details for the seven datasets used as test sets. These datasets cover a diverse range of anomalies across domains such as crime (UCF-Crime, XD-Violence, LAD, MSAD), pedestrian (UBNormal), and traffic (DoTA, LAD), where each test set represents a distinct scenario with its own anomaly definition. There exists many categories that are not included in PreVAD, where some are unseen categories (e.g., *Arrest*), while others appear in PreVAD's normal videos (e.g., *Pedestrian on Road*). This poses a challenge to the open-world generalization ability of the model. Notably, the DoTA dataset only contains abnormal videos, therefore, the predictions on this dataset are normalized for evaluation.

### E.2 PROTOCOLS

We set up two protocols to evaluate the model, where Protocol 1 **comprehensively** evaluates the open-world generalization capability of models, and Protocol 2 **specifically** evaluates robustness to concept drift.

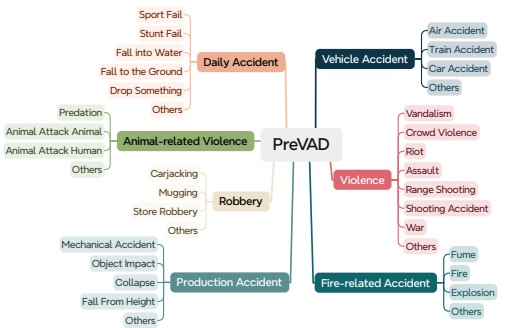

(a) The multi-level taxonomy of PreVAD.

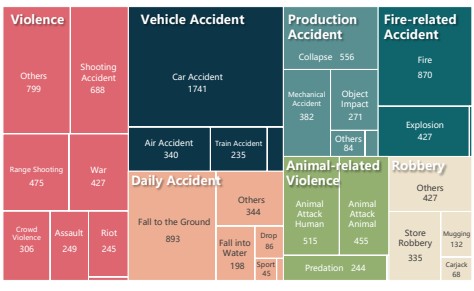

(b) The treemap of PreVAD's category distribution.

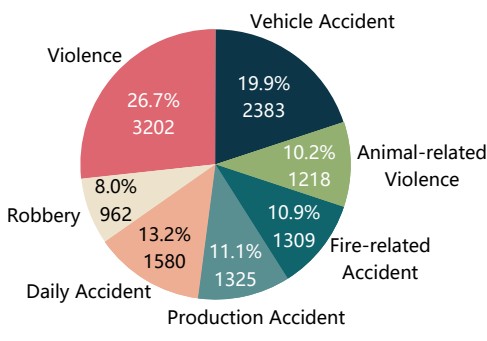

(c) Distribution of coarse-grained categories.

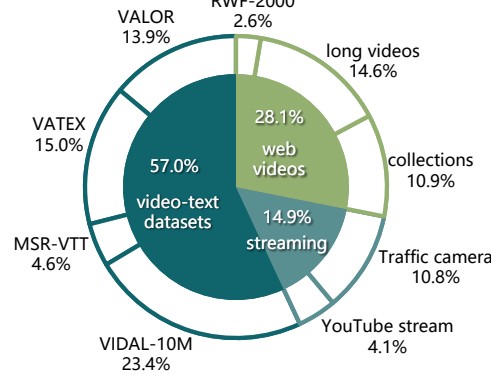

(d) Detailed distribution of video sources.

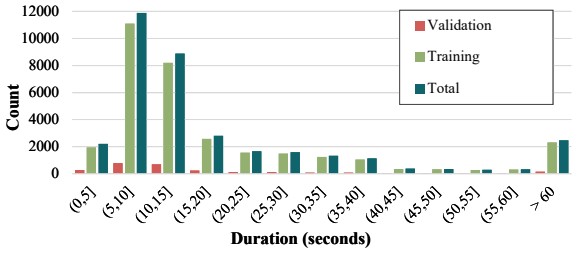

(e) Distribution of the proportion of anomaly in the validation set.

(f) Distribution of the duration.

Figure D: More statistics of PreVAD.

**Protocol 1**  Models are trained on pretraining datasets (e.g., PreVAD) and evaluated in a zero-shot manner across multiple test sets, each representing a distinct scenario with unique definitions of anomalies, video distributions, and anomaly categories. For instance, UCF-Crime focuses on crime-related scenarios and contains normal videos showing pedestrians crossing roads—a behavior considered abnormal in TAD, which focuses on traffic scenarios. Additionally, while UCF-Crime and MSAD primarily consist of surveillance footage, XD-Violence and DoTA include movie clips and dashcam videos, respectively. According to Tab. C, this protocol simulates the actual open-world situation, i.e., training on limited data and testing in different scenarios.

**Protocol 2**  Models are trained on pretraining datasets (e.g., PreVAD) and tested in a zero-shot setting under varying anomaly definitions. Each definition corresponds to a randomly selected subset of anomaly categories from the test set. Videos belonging to the chosen subset are treated as abnormal, while those outside the subset are considered normal. To ensure reliability, we randomly sample five subsets per test set and report the average performance (denoted as drift@5). The selected subsets are detailed in Tab. D. For example, when subset #1 is chosen, videos labeled as fighting, shooting, and riot are regarded as abnormal, whereas videos labeled as normal, abuse, car accident, and ex-

Table C: Overview of test datasets. Abnormal categories not included in PreVAD are underlined: some are unseen categories (e.g., Arrest), while others appear in PreVAD's normal videos (e.g., Pedestrian on Road).

| Datasets | # testing videos | Categories | Notes |
|---|---|---|---|
| UCF-Crime | 290 | Abuse, Arrest, Arson, Assault, Accident, Burglary, Explosion, Fighting, Robbery, Shooting, Stealing, Shoplifting, Vandalism | crime scene from surveillance camera |
| XD-Violence | 800 | Abuse, Car accident, Explosion, Fighting, Riot, Shooting | videos contain shot cut, and part of the data comes from movies |
| MSAD | 240 | Assault, Fighting, People Falling, Robbery, Shooting, Traffic Accident, Vandalism, Explosion, Fire, Object Falling, Water Incident | the accident in the perspective |
| UBNormal | 211 | running, having a seizure, laying down, shuffling, walking drunk, people and car accident, car crash, jumping, fire, smoke, jaywalking, driving outside lane | videos of 3D animation |
| DoTA | 1402 | (9 kinds of fine-grained traffic anomalies) | only contains abnormal videos in both first-person and third-person view |
| TAD | 100 | Accidents, Illegal Turns, Illegal Occupations, Retrograde Motion, Pedestrian on Road, Road Spills, Else | abnormal videos of traffic in both first-person and third-person view |
| LAD | 560 | Crash, Crowd, Destroy, Drop, Falling, Fighting, Fire, Fall Into Water, Hurt, Loitering, Panic, Thiefing, Trampled, Violence | accidents from surveillance and first-person perspectives |

plosion are considered normal. This protocol is specifically designed to assess the ability to handle concept drift issue, which is the main focus of this paper.

Table D: Drift@5 settings for evaluation protocol 2.

| Subset | XD-Violence | MSAD |
|---|---|---|
| 1 | Fighting, Shooting, Riot | Assault, Fighting, Robbery, Shooting, Vandalism, Explosion |
| 2 | Abuse, Car accident, Explosion | People_falling, Traffic_accident, Fire, Object_falling, Water_incident, Explosion |
| 3 | Fighting, Abuse, Explosion | Assault, People_falling, Traffic_accident, Vandalism, Fire, Object_falling |
| 4 | Shooting, Car accident, Riot | Fighting, Robbery, Shooting, Traffic_accident, Water_incident, Fire |
| 5 | Riot, Abuse, Car accident | People_falling, Robbery, Object_falling, Water_incident, Explosion, Assault |

## F  DETAILS OF REPRODUCED METHODS

Since this is a pioneering project, most of our comparisons are based on the results we reproduced with open-source codes and weights. The compared methods can be divided into two groups: non-LLM-based and LLM-based approaches.

**Non-LLM-based methods**  We compare PEL (Pu et al., 2024) and VadCLIP (Wu et al., 2024c), which achieve state-of-the-art performance in video anomaly detection, along with ActionCLIP (Wang et al., 2021) and ViFi-CLIP (Rasheed et al., 2023), SOTA methods in open-vocabulary action recognition. Additionally, a simple CLIP baseline is also included. All methods utilize the same `ViT-B/16` features, except for PEL, which uses I3D features. For VadCLIP, we employ its *A-branch* and apply the same post-processing method as ours to obtain classification results. For the CLIP baseline, we uniformly sampled 8 frames from normal videos and 8 frames from abnormal segments of abnormal videos. Similarity are computed between the visual features and textual prompts constructed from category names. The probabilities from the 8 frames are averaged to produce the final prediction. ActionCLIP and ViFi-CLIP are handled similarly to the CLIP baseline, except that they directly process 8-frame video clips and output final results. Although this comparison setup inherently favors these models (as they do not require temporal localization), experimental results demonstrate that our method still achieves superior performance.

**LLM-based methods**  We compare general-purpose grounding-capable models including Qwen2-VL (Wang et al., 2024a) and Qwen2.5-VL (Bai et al., 2025), the fine-tuned MLLM-based video anomaly detection method HolmesVAU (Zhang et al., 2025), and the multi-VLM ensemble method LAVAD (Zanella et al., 2024). These methods are capable of processing textual prompts, which we use to inject anomaly definition information. Since Qwen2-VL and Qwen2.5-VL cannot output frame-level scores and only predict boundaries, we set the anomaly score of each frame to 0 when no segment is predicted and to 1 for frames within predicted boundaries. HolmesVAU could

Table E: Ablations of hyperparameters in dynamic video synthesis. Following symbols in Algorithm 1, $\delta_m$ denotes the max number of segments, $\theta$ denotes the synthesis probability, and KNN refers to retrieval of K nearest neighbors.

| $\delta_m$ | $\theta$ | KNN | PreVAD | UCF-Crime | XD-Violence |
|---|---|---|---|---|---|
| 5 | 0.7 | ✓ | **69.98** | **81.12** | **74.25** |
| 1 | - | ✓ | 65.73 | 79.18 | 71.41 |
| 3 | 0.7 | ✓ | 68.24 | 80.34 | 67.96 |
| 7 | 0.7 | ✓ | 66.99 | 80.60 | 78.09 |
| 9 | 0.7 | ✓ | 67.13 | 79.28 | 72.90 |
| 5 | 0.7 | | 67.85 | 79.98 | 68.95 |
| 5 | 1 | ✓ | 67.86 | 78.81 | 71.63 |
| 5 | 0.3 | ✓ | 66.53 | 80.50 | 68.74 |

Table F: Ablation on prompting method during inference.

| Prompt method | UCF-Crime AUC |
|---|---|
| class name | 80.44 |
| manual prompt (default) | 81.12 |
| video-specific description | 83.03 |

not be included in the zero-shot comparison on XD-Violence since its training set contains videos from XD-Violence. Although a direct comparison of computational efficiency is challenging due to differences in hardware and implementation optimizations, our proposed method are significantly faster (minutes vs. hours during evaluation).

Table G: Ablation on each component (full version). *guided* refers to guiding the detection with language. *Det. Avg.* refers to the average zero-shot detection performance on seven datasets. *Cls. Avg.* refers to the average zero-shot classification performance on UCF-Crime and XD-Violence.

| $\mathcal{L}_{dvs}$ | $\mathcal{L}_{neg}$ | guided | PreVAD | average metrics | | Detection | | | | | | | Classification | | | |
|---|---|---|---|---|---|---|---|---|---|---|---|---|---|---|---|---|
| | | | | Det. Avg. | Cls. Avg. | UCF | XD | MSAD | UBN | DoTA | TAD | LAD | UCF ACC | UCF F1 | XD ACC | XD F1 |
| ✓ | ✓ | ✓ | 69.98 | 76.42 | 52.57 | 81.12 | 74.25 | 90.41 | 58.07 | 62.60 | 89.56 | 78.91 | 51.72 | 16.64 | 78.13 | 63.80 |
| | ✓ | ✓ | 65.73 | 73.51 | 51.73 | 79.18 | 70.24 | 84.14 | 55.79 | 60.02 | 85.95 | 79.23 | 44.83 | 16.25 | 79.00 | 66.82 |
| ✓ | | ✓ | 68.92 | 73.96 | 51.85 | 77.66 | 70.66 | 90.02 | 57.56 | 59.88 | 83.26 | 78.65 | 48.97 | 17.76 | 75.88 | 64.78 |
| | | ✓ | 67.35 | 71.31 | 48.81 | 77.77 | 62.49 | 89.50 | 55.80 | 57.50 | 80.92 | 75.19 | 43.82 | 17.60 | 73.75 | 60.06 |
| ✓ | ✓ | | 69.87 | 73.84 | 46.23 | 76.64 | 71.39 | 88.60 | 56.88 | 61.40 | 83.96 | 78.02 | 46.21 | 13.28 | 66.87 | 58.54 |

## G  MORE EXPERIMENTAL RESULTS

### G.1  ABLATIONS

In Tab. F, we experiment with different prompting methods: 1) *class name* uses only category labels as anomaly definitions (e.g., explosion, fighting), 2) *manual prompt* employs human-designed description of the category (e.g., *Explosion, often resulting in fire, smoke, and scattered debris*), which serves as our default approach, and 3) *video-specific description* utilizes the description of specific abnormal events as definitions (e.g., *two people catch fire in the explosion near the garage*). Experimental results demonstrated that better prompts could enhance performance. Although video-specific descriptions are unavailable in standard test datasets, it reveals our method's potential for practical applications like locating relevant surveillance clips when specific event details are known.

We provide the complete version of Tab. 5 in Tab. G, which contains more metrics of ablations. We also report hyperparameter search results of dynamic video synthesis in Tab. E. And Tab. E illustrates the importance of large-scale datasets.

### G.2  QUALITATIVE RESULTS

As in Fig. I, we present more zero-shot comparisons on different datasets. The compared three methods all performed visual-language alignment. VadCLIP and LaGoVAD, which utilize pre-

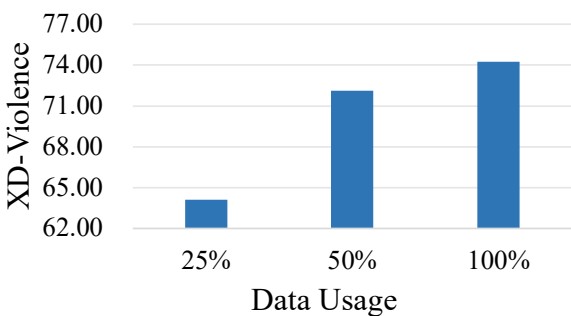

Figure E: Ablations of data usage during training.

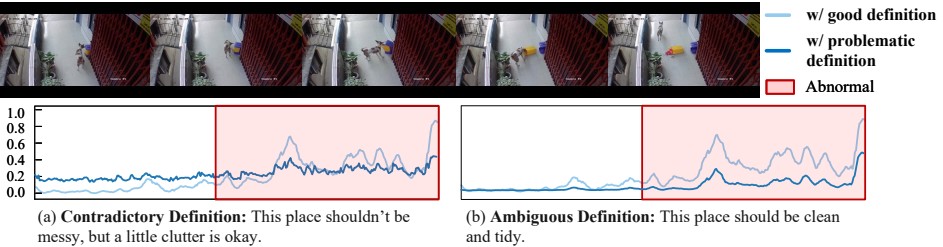

(a) **Contradictory Definition:** This place shouldn't be messy, but a little clutter is okay.

(b) **Ambiguous Definition:** This place should be clean and tidy.

Figure F: Visualization of failure cases under contradictory and ambiguous definitions. The *good* definition is consistent with the prompt shown in Figure 4(b), i.e., *Knocking over a trashcan is abnormal*.

aligned feature extractors, demonstrate a significant advantage in cross-dataset testing, further proving the importance of semantic alignment for generalization. Additionally, our proposed LaGoVAD achieves predictions with fewer false alarms, thanks to the contrastive learning with in-sample negative mining we conducted. Finally, LaGoVAD was also able to make accurate predictions for the novel *Arrest* category.

Fig. J shows a prediction example of LaGoVAD on the XD-Violence dataset. The primary anomaly in this video is a riot scene, accompanied by secondary anomalies including fighting, explosion, and shooting. We visualize the outputs from both the detection head and the classification head. Our model achieves accurate detection in normal intervals (a-c) as well as in riot scenarios (b). Furthermore, the model effectively captures mixed fighting (e) and shooting events (f-g) through its classification head, demonstrating a sensitive response to co-occurring anomalies. Notably, even for transient explosion events (d) with extremely short durations, our model could produce sharp response spikes.

We provide some failure cases of the proposed method in Fig. F. When the definition is internally contradictory, LaGoVAD produces unstable and unreliable scores because it has no mechanism of handling contradiction (e.g., request clarification from the user). As for ambiguous definition, LaGoVAD shows a certain level of robustness, which we attribute to the semantic understanding capability of the CLIP text encoder.

## H   DISCUSSION OF THE ASSUMPTION

This section discusses some special cases in Assumption 1.

One scenario arises when additional hidden factors influence abnormality beyond the explicit definition. For example, consider an access-control rule where entering a corridor is abnormal only during examination periods. If the examination schedule is not part of the definition, it would affects whether a behavior is abnormal. However, such factors can be incorporated into an augmented anomaly definition $Z$, and the assumption remains valid once the definition is expanded to include these external conditions.

Another scenario occurs when the anomaly definition is entirely dependent on the video. However, many anomaly definitions in practice are not inferable from the visual evidence alone. For instance, if it is impossible to determine from the scene whether smoking is permitted, the abnormality of smoking depends on an external policy rather than the video itself. Similarly, carrying a backpack may be abnormal only in restricted zones, but if the boundary of the restricted zone is not visually indicated, the definition cannot be recovered from the video. These cases further demonstrate the validity of our assumption.

## I  PROOF OF PROPOSITIONS

*Proof of Proposition 1.* Under Assumption 1, there exists a deterministic function $\mathcal{F}$ such that whenever $(V = v, Z = z, Y = y)$ occurs with positive probability, there exists $y = \mathcal{F}(v, z)$.

We now compute the conditional distribution $P_d(Y = y \mid V = v, Z = z)$ under any domain $d$. By the definition of conditional probability,

$$P_d(Y = y \mid V = v, Z = z) = \frac{P_d(V = v, Z = z, Y = y)}{P_d(V = v, Z = z)}. \tag{15}$$

Because of Assumption 1, for any fixed $(v, z)$, there is exactly one value $y^* = \mathcal{F}(v, z)$ such that

$$P_d(V = v, Z = z, Y = y^*) = P_d(V = v, Z = z), \tag{16}$$

and for all $y \neq y^*$,

$$P_d(V = v, Z = z, Y = y) = 0. \tag{17}$$

Substituting these two cases into the conditional probability formula:

$$P_d(Y = y \mid V = v, Z = z) = \begin{cases} 1, & y = \mathcal{F}(v, z), \\ 0, & y \neq \mathcal{F}(v, z). \end{cases} \tag{18}$$

This distribution depends only on the deterministic mapping $y = \mathcal{F}(v, z)$ and does not depend on the choice of domain $d$. Therefore, for any two domains $d_1$ and $d_2$,

$$P_{d_1}(Y \mid V, Z) = P_{d_2}(Y \mid V, Z). \tag{19}$$

□

*Proof of Proposition 2.* By the law of total probability, for any domain $d$, video $v$ and label $y$, we have

$$P_d(Y = y \mid V = v) = \sum_z P_d(Y = y \mid Z = z, V = v) \, P_d(Z = z \mid V = v). \tag{20}$$

Under Assumption 1, the label $Y$ is deterministically given by $\mathcal{F}(V, Z)$. Hence, for all $y \neq \mathcal{F}(v, z)$, we have $P_d(V = v, Z = z, Y = y) = 0$, which implies

$$P_d(Y = y \mid V = v) = \sum_{z : \mathcal{F}(v, z) = y} P_d(Z = z \mid V = v). \tag{21}$$

Now assume that there exists at least one pairs of $v^\star, y^\star$ such that

$$\sum_{z : \mathcal{F}(v^\star, z) = y^\star} P_{d_1}(Z = z \mid V = v^\star) \neq \sum_{z : \mathcal{F}(v^\star, z) = y^\star} P_{d_2}(Z = z \mid V = v^\star), \tag{22}$$

applying this identity to Eq. 21 yields

$$\sum_{z : \mathcal{F}(v^\star, z) = y^\star} P_{d_1}(Z = z \mid V = v^\star) \neq \sum_{z : \mathcal{F}(v^\star, z) = y^\star} P_{d_2}(Z = z \mid V = v^\star). \tag{23}$$

Therefore,

$$P_{d_1}(Y \mid V) \neq P_{d_2}(Y \mid V). \tag{24}$$

□

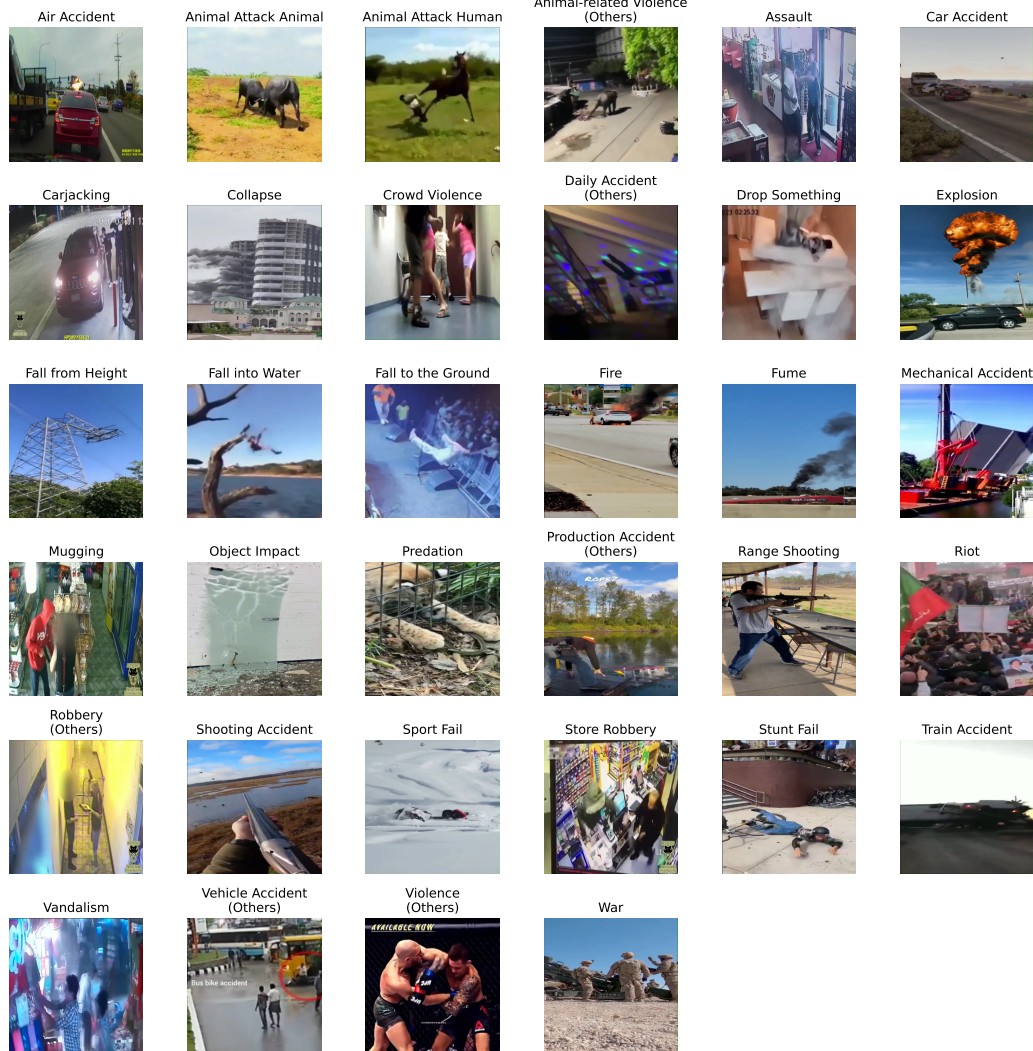

Figure G: Examples of each abnormal category in PreVAD.

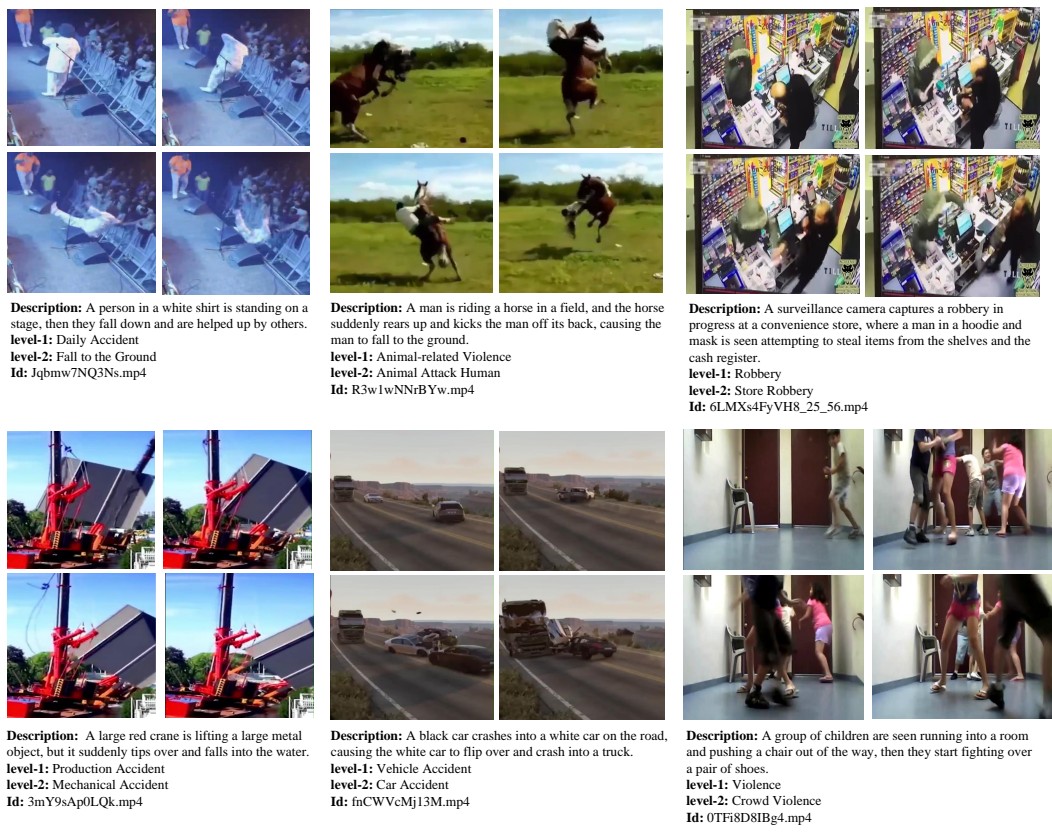

Figure H: Examples of annotations in PreVAD.

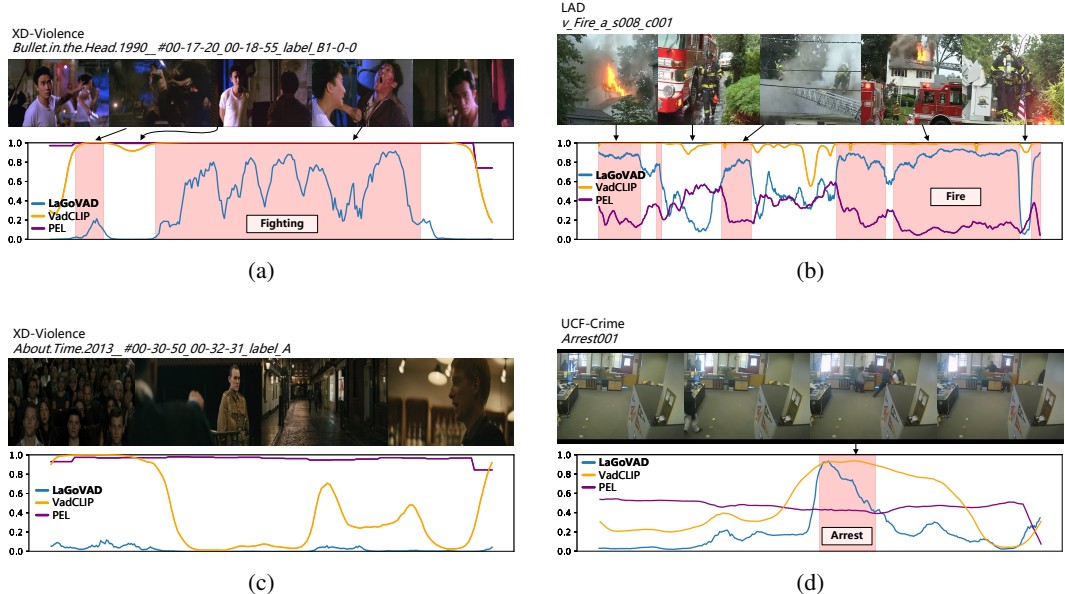

Figure I: Qualitative comparisons with other models. The two models used for comparison are trained on the UCF dataset (a)(b)(c) or the XD-Violence dataset (d). All the results are zero-shot results.

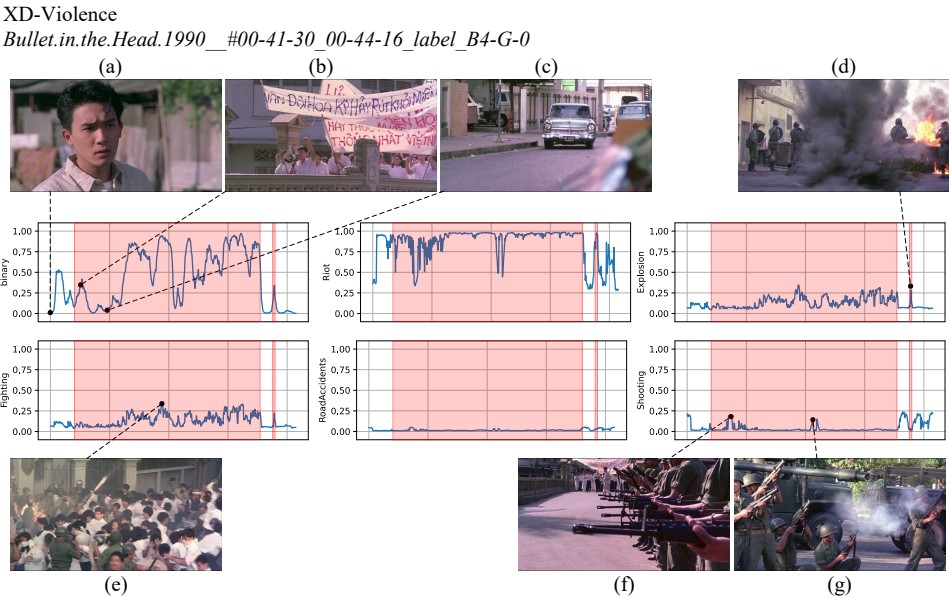

Figure J: Visualization of results of the detection head and the classification head of the proposed LaGoVAD. The top-left plot illustrates the detection head output, while the remaining plots correspond to the classification head outputs, activated via the Sigmoid function. Frames (a)-(g) denote key frames. **LaGoVAD is able to detect fine-grained anomalies in a video that contains multiple different anomalies.**

