# OpenReview forum: "Language-guided Open-world Video Anomaly Detection under Weak Supervision"
_ICLR.cc/2026/Conference — ICLR 2026 Poster_

### Official Review · Reviewer_vsXK · 2025-10-29

**Soundness:** 2
**Presentation:** 3
**Contribution:** 2
**Rating:** 6
**Confidence:** 2

**Summary:**

This paper proposes LaGoVAD, a framework for video anomaly detection (VAD) in open-world scenarios, where anomaly definitions can change dynamically according to user-specified natural language input. Also, the authors introduce PreVAD, a dataset with multi-level anomaly categories and textual descriptions.

**Strengths:**

- Originality: The paper reformulates VAD as a mapping Φ:(V, Z)→Y, explicitly modeling dynamic anomaly definitions to address concept drift.

- Quality: Extensive zero-shot and drift@5 protocols on seven datasets demonstrate SOTA results across both detection and classification metrics (Tabs. 2–4).

- Clarity: The paper is well organized.

- Significance: Dynamic video synthesis and contrastive loss with hard negative mining jointly improve generalization under weak supervision. PreVAD introduced by this paper is a large and semantically rich VAD dataset with 35,279 videos and detailed anomaly descriptions. Users can specify anomalies in natural language, allowing adaptive definitions at inference time.

**Weaknesses:**

- Limited theoretical justification of concept-drift handling. While Eq. (4) formalizes the dependency on Z, there is no quantitative analysis showing explicit mitigation of Ptrain ≠ Ptest beyond empirical gains.
- The cross-modal analysis is weak. The fusion mechanism is relatively simple, and its contribution is not isolated in ablations beyond $L$dvs / $L$neg.
- The dataset annotation reliability is questionable. No quality metrics are presented.

**Questions:**

- Can authors provide with quantitative analysis of concept-drift mitigation?
- Can authors discuss and test failure cases where user-provided definitions are ambiguous or contradictory?
- Can LaGoVAD work in real-time video streams where the user updates or changes the anomaly definition while the system is running?

**Details Of Ethics Concerns:**

- The paper does not specify any anonymization, consent, or privacy-preserving mechanisms in dataset collection or model deployment.

- The Ethics Statement in the paper acknowledges possible misuse but lacks detailed mitigation guidelines.

- The PreVAD dataset includes human-labeled and MLLM-generated annotations for video anomalies but provides no details about annotator compensation or consent.

---

> ### Author Response · Authors · 2025-11-21
>
> **W1 & Q1 (Theoretical justification):**
>
> Our core assumption is that $Y$ is determined solely by $Z$ and $V$, meaning the anomaly decision depends only on the video and the definition itself. Under this assumption, we obtain $P_{train}(Y \mid V) \neq P_{test}(Y \mid V)$ and $P_{train}(Y \mid V, Z) = P_{test}(Y \mid V, Z)$, which we prove in Section I. Since for any given video and a specific definition, its abnormality remains consistent across training and testing. Modeling $\Phi:(V,Z)\rightarrow Y$ therefore provides the necessary condition for avoiding concept drift. Our experiments, particularly under Protocol 2, further provide quantitative analysis that the proposed formulation effectively mitigates concept drift.
>
> We have revised the Section 3 for better presentation and provided the theoretical proof in Section I.
>
>
>
> **W2 (Cross-modal module):**
>
> This work focuses on handling the core challenge of concept drift and open-world. Our main contributions include a novel paradigm to avoid concept drift, two architecturally independent regularization strategies, and a large-scale dataset. Therefore, we will analyze and improve the fusion mechanism in the future work.
>
>
>
> **W3 (Annotation reliability):**
>
> Thank you for the concern. For category-level annotations, all annotators had domain expertise in video anomaly detection, and a review process was applied to samples with inconsistent labels. For description annotations, we conducted a subjective quality assessment of automatically generated texts in the early-stage. Three annotators independently scored descriptions produced with and without constrained prompts. We randomly sampled 200 videos, and asked each annotator to rate on a 1–5 scale.
>
> On average, the descriptions generated *without* constrained prompts received a score of 3.5, while those generated *with* constrained prompts received a score of 4.3, demonstrating the effectiveness of the constrained prompt. We also assess the description annotations of HIVAU-70K dataset used by Holmes-VAU with identical criteria, which receive a score of 4.4, indicating the noise level in our annotation is consistent with that of other works. We also calculated the intraclass correlation coefficient (ICC) [i] to quantify inter-annotator agreement. The ICC(2,k) value was 0.799, indicating a high level of consistency across annotators.
>
> We have provided more details in Section D.2.
>
>
> **Q2 (Ambiguous definition):**
>
> Thank you for the suggestion. In the revised version, we have added an analysis of failure cases in the appendix (Section G.2). Interestingly, we found that LaGoVAD shows a certain level of robustness to ambiguous user-provided definitions, which we attribute to the semantic understanding capability of the CLIP text encoder. However, when the definition is internally contradictory, LaGoVAD has no mechanism to request clarification from the user and therefore produces unstable and unreliable scores.
>
>
> **Q3 (Real-time):**
>
> Yes. Users can update the anomaly definition at any time, and each update only requires a single forward pass through the CLIP text encoder, which is computationally negligible. Since inference does not rely on LLMs or MLLMs, LaGoVAD runs efficiently and is well suited for real-time scenarios.
>
>
> **Ethics Concerns:**
>
> Thank you for highlighting these important ethical considerations. PreVAD is constructed entirely from publicly available sources commonly used in prior video understanding research [ii, iii]. No personal data, identities, or sensitive attributes were collected. There are no privacy issues involved in the model deployment. We will impose license restrictions on commercial use to prevent misuse. The annotators all participated voluntarily and were paid at a rate approximately twice the local minimum wage.
>
>
>
> [i] Shrout, Patrick E., and Joseph L. Fleiss. "Intraclass correlations: uses in assessing rater reliability." *Psychological bulletin* 86.2 (1979): 420.
>
> [ii] Xu, Jun, et al. "Msr-vtt: A large video description dataset for bridging video and language." *Proceedings of the IEEE conference on computer vision and pattern recognition*. 2016.
>
> [iii] Wang, Xin, et al. "Vatex: A large-scale, high-quality multilingual dataset for video-and-language research." *Proceedings of the IEEE/CVF international conference on computer vision*. 2019.

---

> > ### Comment · Area_Chair_tJ24 · 2025-11-26
> > **Ethics concerns**
> >
> > Dear Authors,
> >
> > As pointed out by **Reviewer vsXK**, there may be major ethics concerns in the proposed VAD dataset. We appreciate the response to these concerns, but it is mismatched to the observations of the AC (and the reviewers). For example, the videos collected include those from i) existing video-text datasets (Xu et al., 2016; Wang et al., 2019; Liu et al., 2025; Zhu et al., 2024a), ii) web resources, and iii) YouTube streams and traffic camera streams. As shown by exemplar video frames in the paper, sensitive data—such as human faces, vehicle registration number on car plates, etc—are clearly visible. It is unclear whether and how the authors anonymized the video data. Please elaborate in detail on whether the dataset has these issues, and how they have been properly addressed.
> >
> > Thank you!
> >
> > Your AC, ICLR 2026

---

> > > ### Author Response · Authors · 2025-11-26
> > > **Response to AC**
> > >
> > > Dear AC,
> > >
> > > Thank you very much for raising the potential ethical concerns in our dataset. Below, we elaborate on each point:
> > >
> > > i) Regarding existing video–text datasets, we will release the IDs or URLs of the videos used in PreVAD. As these videos are part of previously published and properly licensed datasets, and we only provide IDs/URLs instead of redistributing raw video files, **we confirm that this portion raises no ethical or privacy concerns.**
> > >
> > > ii) For newly collected web videos, similar to existing datasets [i, ii], we will release the publicly accessible video URLs along with the time ranges used for sampling. Since we do not release or host any original video files, and all sampled videos are publicly available online, **we confirm that this part also raises no ethical or privacy concerns.**
> > >
> > > iii) For data recorded from live streams, we will provide multiple **anonymized features** (e.g., CLIP embeddings) to support research usage without exposing personal information. When releasing the raw videos, we will **apply face detection and license plate detection to blur sensitive regions**, following an anonymization pipeline consistent with prior works [iii].
> > >
> > > Additionally, regarding the dataset examples shown in the paper, Figures 3(c), A, C, G, and H include samples from our dataset. We believe that none of these images contain faces or license plates that can clearly reveal personal identity. However, if any example is still considered unsuitable for publication, we are willing to apply further masking or removal. For Figures I and J, which contain clear faces, these samples come from external datasets [iv] and are part of movie scenes, which we believe do not raise privacy concerns.
> > >
> > > Thank you again for your valuable feedback. We agree that ethics and privacy are core to dataset quality, and we will ensure the final release fully follows proper anonymization standards.
> > >
> > > Sincerely,
> > >
> > > The Authors
> > >
> > >
> > >
> > > [i] Xu, Jun, et al. "Msr-vtt: A large video description dataset for bridging video and language." *Proceedings of the IEEE conference on computer vision and pattern recognition*. 2016.
> > >
> > > [ii] Wang, Xin, et al. "Vatex: A large-scale, high-quality multilingual dataset for video-and-language research." *Proceedings of the IEEE/CVF international conference on computer vision*. 2019.
> > >
> > > [iii] Cao, Congqi et al. “A New Comprehensive Benchmark for Semi-supervised Video Anomaly Detection and Anticipation.” *2023 IEEE/CVF Conference on Computer Vision and Pattern Recognition (CVPR)* (2023): 20392-20401.
> > >
> > > [iv] Wu, Peng et al. “Not only Look, but also Listen: Learning Multimodal Violence Detection under Weak Supervision.” *ArXiv* abs/2007.04687 (2020).

---

> > ### Comment · Reviewer_vsXK · 2025-11-27
> >
> > I thank the authors for the detailed response and the revisions. The discussion on annotation reliability is much clearer now, the subjective scoring and inter-annotator agreement address that concern well. The analysis of ambiguous or contradictory definitions and the explanation of real-time capability is also clear and convincing. In the revised version, the authors expand the theoretical section with formal propositions and proofs and give a better motivation for modeling the definition as a stochastic variable. However, the paper still doesn’t offer an empirical analysis that isolates the effect of this formulation. The response notes that fusion isn’t the main focus and that improvements are left for future work, but that doesn’t fully resolve the original weakness. Overall I’m satisfied with the authors’ response.

---

> > > ### Comment · Area_Chair_tJ24 · 2025-11-27
> > >
> > > Dear Authors,
> > >
> > > Thanks for the response. Please further clarify the following two points:
> > > - If only the URLs are provided for most of the raw data, would these URLs become invalid in future? how would you guarantee a long-term accessibility to the dataset?
> > > - It is mentioned in the response that you "***will** apply face detection and license plate detection to blur sensitive regions*".  How would the blurring affect the performance? and further, how would this change affect the arguments on the empirical performance in the current paper version?
> > >
> > > The Ethics Chairs may check whether the provision of the URLs involves ethical/privacy issues.
> > >
> > > Thank you!
> > >
> > > AC, ICLR 2026

---

> > > > ### Author Response · Authors · 2025-11-27
> > > > **Response to AC**
> > > >
> > > > Dear AC,
> > > >
> > > > Thank you for your attention. Our clarification is as follows:
> > > >
> > > > - Most current web video datasets are released in the form of URLs, and it is inevitable that after some time, a portion of videos may become unavailable. For example, based on the authors’ experience, 10% of the VATEX dataset become unavailable after 4–5 years. Under this proportion, and referring to the trend shown in Figure E, the impact on performance can be ignored. Therefore, this proportion is acceptable for large-scale datasets. In addition, we will provide the extracted features as an alternative backup.
> > > >
> > > > - First, in the VAD task, the abnormal patterns are primarily reflected in motion (e.g., falling) or sudden scene changes (e.g., explosions). There is no evidence that applying blurring will have a significant impact on the model. Second, for the 15% live stream data, we can confirm that the traffic camera subset, which accounts for 10.8% of the entire dataset, has already been processed and contains almost no faces or clearly identifiable license plates or other sensitive privacy information. Only the live streaming recordings, which account for 4.1% of the entire dataset, include a small portion of videos that require additional processing. Finally, some works [i, ii] have confirmed that blurring operations have almost no impact on tasks that do not rely on sensitive data.
> > > >
> > > > As other reviewers have noted, *PreVAD is a significant contribution to the community* and *I strongly encourage the authors to release the dataset*. We appreciate the AC’s efforts in ensuring that the dataset release follows common practices and complies with the ICLR Code of Ethics.
> > > >
> > > > Sincerely,
> > > >
> > > > The Authors
> > > >
> > > > [i] Hukkelås, Håkon and Frank Lindseth. “Does Image Anonymization Impact Computer Vision Training?” CVPRW (2023): 140-150.
> > > >
> > > > [ii] Yang, Kaiyu et al. “A Study of Face Obfuscation in ImageNet.” ICML (2021).

---

> > > ### Author Response · Authors · 2025-12-02
> > > **Response to the two follow-up points raised by Reviewer vsXK**
> > >
> > > **Empirical analysis of formulation**
> > >
> > > Thank you for your recognition of our formulation. In the first column of Table 5, we present an ablation study on whether a dynamic anomaly language definition is used, which directly corresponds to our proposed stochastic formulation. The results show that when this formulation is disabled, the in-domain performance does not change significantly, but the cross-domain performance drops dramatically, due to the model’s inability to handle concept drift in the anomaly definition.
> > >
> > > **Modal fusion mechanism**
> > >
> > > We appreciate the reviewer’s feedback. However, we respectfully maintain that modality fusion analysis is not a *weakness* of the current submission, but rather *outside its primary focus*. The central goal of this work is modeling the dynamic anomaly definition, rather than to propose a new multi-modal fusion architecture. In this paper, we employ one of the most commonly used fusion architecture in the era of Transformers without specialized tricks, ensuring that the overall framework remains generic.

---

### Official Review · Reviewer_BtGb · 2025-10-29

**Soundness:** 2
**Presentation:** 3
**Contribution:** 2
**Rating:** 6
**Confidence:** 4

**Summary:**

This paper introduces a language-guided open-world video anomaly detection framework that dynamically adapts to user-defined anomalies via natural language. The authors propose LaGoVAD, a model incorporating dynamic video synthesis and contrastive learning with hard negative mining, and contribute PreVAD—a large-scale dataset with semantic anomaly descriptions.

**Strengths:**

- Novel formulation of concept drift as a conditional modeling problem.
- Well-designed regularization strategies to combat overfitting.
- Comprehensive evaluation across 7 datasets with a dedicated concept drift protocol.
- PreVAD is a significant contribution to the community.

**Weaknesses:**

1. **Limited Evaluation of Prompt Robustness**: The model is only tested with manually curated, category-level prompts. Its performance under noisy, ambiguous, or open-domain user descriptions—common in real-world use—remains unverified.

2. **Semantic Consistency in Video Synthesis**: The dynamic video synthesis module relies solely on CLIP-based global features for segment retrieval. This may lead to semantically inconsistent videos under domain shifts (e.g., real-world vs. synthetic data), potentially undermining the regularization effect. The paper lacks analysis of synthesis quality or failure cases in cross-domain settings.

3. **Risk of Noisy Hard Negatives**: The contrastive learning strategy samples hard negatives from normal segments within anomaly videos. However, in cases of ambiguous normal-abnormal boundaries (e.g., “pedestrian on road”), these segments may be mislabeled, introducing noise into the contrastive objective. The impact of such false negatives is not discussed.

4. **Architectural Simplicity**: While the model effectively instantiates the proposed paradigm, its architecture remains relatively simple. More sophisticated multimodal fusion mechanisms or temporal modeling approaches could further enhance performance.

**Questions:**

1. How does the contrastive learning module handle potential mislabeling of hard negatives in ambiguous scenarios?
2. Has any prompt normalization or robustness strategy been considered to handle diverse or noisy user inputs?
3. Could finer-grained retrieval mechanisms (e.g., action-based or scene-graph matching) improve semantic coherence in synthesized videos under domain shift?

---

> ### Author Response · Authors · 2025-11-21
>
> **W1 & Q2 (Limited Evaluation of Prompt Robustness):**
>
> Thank you for the insightful suggestion. To the best of our knowledge, this is among the first works to systematically examine concept drift in open-world video anomaly detection, and thus our primary goal was to establish an effective baseline. In this work, we do not apply explicit prompt normalization or robustness strategies, but CLIP text encoder inherently presents robustness towards prompts. We have included corresponding examples in the revised version. In the future, we plan to leverage LLMs to better infer and interpret users’ real intentions in practical scenarios.
>
>
> **W2 & Q3 (Semantic Consistency in Video Synthesis):**
>
> Thank you for raising this thoughtful concern. The visual features used in our dynamic video synthesis module are the same features fed into the Temporal Encoder. As long as two segments exhibit high CLIP similarity, they are largely indistinguishable from the model’s perspective. Therefore, even if two segments may appear semantically discontinuous to human observers, this does not undermine the regularization effect for the model.
>
> What truly affects the regularization effect is whether the retrieval step can find sufficiently similar segments. In our statistics, the retrieved clips used for synthesis have an average similarity of 87.2%, which in our experiments is sufficiently high for the intended purpose. Moreover, This mixture of real and synthesized samples further alleviates potential domain shift between real-world and synthetic segments.
>
> We have added the clarification in the revised version (Section 4.1).
>
>
> **W3 & Q1 (Risk of Noisy Hard Negatives):**
>
> Thank you for pointing this out. Due to the inherent limitations of weak supervision, perfectly separating positives and negatives is generally infeasible, so some level of noise is expected. However, this noise can be regulated through the temperature coefficient $\eta$ in Eq.~(9). A lower temperature produces a sharper distribution, reducing the likelihood of selecting ambiguous samples (though resulting in easier negatives), while a higher temperature introduces harder but potentially noisier negatives. Empirically, we set $\eta = 0.02$ to strike a balance between hardness and noise, and did not observe noticeable training instability.
>
>
>
> **W4 (Architectural Simplicity):**
>
> Thank you for the suggestion. We intentionally adopt a simple baseline as our main focus is handling the challenge of concept drift and the open world. Our experiments have shown that even such a simple structure can achieve significant improvements, which further demonstrates the effectiveness of the proposed paradigm. Moreover, the proposed two regularizers can be easily integrated with more sophisticated fusion and temporal modeling modules, a direction we plan to explore in the future.

---

### Official Review · Reviewer_2UdC · 2025-10-29

**Soundness:** 3
**Presentation:** 4
**Contribution:** 3
**Rating:** 4
**Confidence:** 4

**Summary:**

This paper addresses the challenge of open-world video anomaly detection (VAD) under weak supervision, focusing on the critical and practical issue of concept drift—where the definition of anomalies evolves over time and across scenes. The main contributions include a newly proposed large-scale video anomaly dataset and two novel training strategies. The dataset is characterized by its use of language-based, variable definitions of anomalies. The authors also introduce two training techniques: dynamic video synthesis and contrastive learning with negative mining.

**Strengths:**

1. The paper is well-written and easy to follow.
2. The problem addressed is both interesting and important in the field of VAD.
3. The experiments are comprehensive and the zero-shot performance demonstrated is promising.

**Weaknesses:**

1. The evaluation under Protocol 2 is incomplete. As currently designed, Protocol 2 only considers concept drift in one direction—where abnormal events in the pre-training dataset (PreVAD) may shift to normal. The reverse scenario, in which normal events become abnormal, is not evaluated. It would be beneficial to include this additional evaluation.
2. While the two proposed regularization strategies are technically sound, both video splicing and hard negative mining have been extensively explored in prior work. More importantly, the connection between these strategies and the issue of concept drift, which is a key characteristic of the proposed datase, remains unclear.
3. The performance improvements are not substantial. As shown in Table 4, LaGoVAD only achieves marginal gains over VadCLIP in terms of AP on XD-drift@5 and AUC on MSAD-drift@5.

**Questions:**

1. What training set was used for VadCLIP in Table 4? Was VadCLIP pre-trained on XD-violence or PreVAD?
2. Did you experiment with textual prompt ensembling during inference? How does the performance differ when using class names versus anomaly descriptions?

---

> ### Author Response · Authors · 2025-11-21
>
> **W1 (Incomplete protocol2):**
>
> Thank you for raising this concern. While the comment focuses on the relation between PreVAD and downstream subsets, Protocol 2 is actually defined between subsets, rather than between PreVAD and any specific subset, i.e., $P_{subset\_i}(Y|V) \neq P_{subset\_j}(Y|V), i \neq j$. Therefore, it considers both directions. For example, from subset-1 to subset-2 of the MSAD dataset , *Vandalism* shifts from abnormal to normal, while *Water\_incident* shifts from normal to abnormal.
>
> This zero-shot evaluation is designed not impose additional restrictions on the choice of training data, as it is intended to reflect performance under varying user requirements / anomaly definitions. For instance, Qwen2.5-VL and LAVAD are trained on general-purpose data, whereas HolmesVAU is trained on UCF-Crime and XD-Violence.
>
> Moreover, although PreVAD pre-defines a set of categories, our model does not statically treat any video/event as abnormal during training. Our training samples are randomly drawn from the joint distribution $P(V, Z, Y)$. Each sampled triplet $ (v, z, y) $ may represent, for example:
>
> - $ (v=\text{falling video},\ z=\text{“falling” is abnormal},\ y=\text{video is abnormal}) $,
>
> - $ (v=\text{falling video},\ z=\text{“falling” is normal},\ y=\text{video is normal}) $.
>
> Therefore, $P_{train}(Y|V)$ is inherently variable during training in our method, so the model is not tied to a fixed abnormal/normal assignment for any event, which naturally aligns with Protocol 2.
>
> Clarification is added in the Section 6.1 of the revised version.
>
>
> **W2 (Regularization novelty & motivation):**
>
> Thank you for the insightful comment. Existing video splicing and hard negative mining strategies discussed in the broader literature cannot be directly applied to the proposed open-world VAD framework. Most prior approaches either enhance spatial localization through cut-and-paste or mix-up strategies, or improve temporal grounding under fully supervised settings. In contrast, our Dynamic Video Synthesis module performs semantics-aware duration augmentation to strengthen temporal localization, while our Hard Negative Mining operates on weakly labeled videos and conducts contrastive learning under dynamic anomaly definitions.
>
> Both regularization strategies are motivated by the increased complexity of the multimodal space induced by modeling $\Phi: (V, Z) \rightarrow Y$. When anomaly definitions $Z$ change, the sample density around each definition becomes sparse, which makes the model more prone to overfitting. Dynamic Video Synthesis increases the diversity and coverage of temporal patterns, while Hard Negative Mining increases the sample's quality. These two regularizers work together with other proposed major components (dataset, training paradigm) to achieve open-world generalization.
>
> We have revised our introduction section and related work section to better position our regularization strategies. If there are specific publications the reviewer has in mind, we would be glad to include and discuss them.
>
>
> **W3 (Performance):**
>
> In Protocol 2, both AUC and AP are valid metrics and should be considered jointly. When averaged, our method achieves a 12.4% improvement over VadCLIP. In addition, Protocol 1 (Tables 2 and 3) also serves as our major comparison for comprehensively evaluating open-world generalizability, where we observe significant and consistent performance gains.
>
>
> **Q1 (Details in Tab.4):** We apologize for the lack of clarity. VadCLIP reported in Table 4 was trained on the PreVAD. We have added this infomation in Tab. 4.
>
>
> **Q2 (Prompt):**
>
> We did not use prompt ensembling during inference, as doing so may introduce unfairness in comparison with other methods.
>
> Regarding the choice between class names and anomaly descriptions, Table F in the appendix shows that more detailed and specific textual descriptions lead to better performance. In practical applications, users often have prior expectations about the anomalies they care about. For example, a camera placed near the stairs in a nursing home may be configured with a prompt such as “an elderly person slipping on the stairs.” Such a level of prompt engineering is both realistic and feasible in real-world deployments.

---

> > ### Comment · Reviewer_2UdC · 2025-11-28
> > **Responses**
> >
> > I appreciate the authors' detailed clarification regarding Protocol 2, which has improved my understanding of the experimental design. The rationale behind the proposed regularization strategies - addressing the scarcity of concept-shifted anomalies while preventing overfitting through data augmentation - is clearly presented. However, the study would benefit from a more thorough discussion of the fundamental challenges associated with concept shift, particularly catastrophic forgetting and task ambiguity, which are also central issues in continual learning. Therefore, I would recommend that the authors could include relevant discussion connecting their work to the continual/life-long learning literature. Additionally, the simple averaging two distinct metrics (AUC and AP) is not meaningful due to their differing interpretations and sensitivities. However, given the novelty and practical significance of this anomaly detection framework, I would revise my score to 6.

---

> > > ### Author Response · Authors · 2025-12-02
> > > **Response to the two follow-up points raised by Reviewer 2UdC**
> > >
> > > **Discussion of continual learning**
> > >
> > > Thank you for the suggestion. Continual learning indeed represents **another** line of research for addressing concept shift. However, to the best of our knowledge, existing VAD approaches within the continual/life-long learning literature do not directly target the concept drift, making a meaningful empirical comparison difficult. While we acknowledge the relevance of this direction, our focus in this work is not on continual-learning techniques, and integrating them is left for future exploration.
> > >
> > > **Averaging two metrics**
> > >
> > > Thank you for pointing this out. Our intention is to emphasize that both metrics capture complementary aspects of detection performance, and the averaged score is used solely as a coarse summary indicator. The full unaveraged metrics are reported in Table 2, 3, and 4.

---

### Official Review · Reviewer_yCS1 · 2025-10-31

**Soundness:** 3
**Presentation:** 3
**Contribution:** 2
**Rating:** 6
**Confidence:** 4

**Summary:**

This paper proposes a language-guided open-world video anomaly detection framework named  LaGoVAD, which enables models to adapt to dynamically changing definitions of anomaly. Unlike traditional VAD methods that assume fixed anomaly categories, LaGoVAD conditions detection on both video content and user-provided textual definitions, effectively addressing the problem of concept drift. The model employs two key strategies—dynamic video synthesis for data diversification and contrastive learning with hard negative mining for robust cross-modal alignment. To support training, the authors introduce PreVAD, a large-scale dataset with 35K videos and fine-grained textual annotations. Extensive zero-shot experiments across seven datasets show that LaGoVAD achieves state-of-the-art performance in both cross-domain and dynamic definition scenarios, advancing the field toward user-controllable, open-world anomaly detection.

**Strengths:**

1. The paper makes a significant contribution to the field of video anomaly detection by addressing one of its fundamental challenges — the context-dependent and evolving nature of anomaly definitions in real-world, open-world scenarios. Instead of assuming a fixed anomaly category, the authors propose a novel language-guided paradigm that allows dynamic redefinition of anomalies through natural language, effectively tackling the long-standing problem of concept drift.
2. This paper introduces the PreVAD, a large-scale, multi-domain dataset with rich textual annotations and hierarchical labeling. This dataset fills an important gap in current VAD research, where existing datasets are often limited in scale, diversity, and semantic granularity.
3. The proposed baseline model, LaGoVAD, is well-designed and demonstrates strong empirical performance on both the new dataset and multiple public benchmarks. Its integration of dynamic video synthesis and contrastive learning with hard negative mining represents a thoughtful and effective combination that improves cross-modal alignment and temporal robustness.

**Weaknesses:**

1. In the current era of Vision-Language Models, the problem of definition drift can often be effectively mitigated by prompt engineering and leveraging pretrained multimodal knowledge from large models. Recent training-free approaches such as LAVAD, SUVAD, and VERA have already explored similar directions, showing that dynamically redefining anomalies through prompts is feasible without additional training. Although the authors explicitly avoid using large language models, the deliberate omission or insufficient discussion of these related works is problematic and weakens the positioning of this paper within the existing research landscape.

2. The proposed framework, while conceptually sound, remains relatively simple in design. Its core components—language conditioning, dynamic video synthesis, and contrastive alignment—are incremental extensions of existing VAD or multimodal alignment methods rather than substantial breakthroughs. As a result, the framework provides only limited innovation in addressing the core challenge of definition drift, offering more of an engineering adaptation than a novel theoretical advancement.

3. The paper claims that modeling the mapping $\Phi :(V,Z)\rightarrow Y$ can “effectively avoid concept drift,” but provides no theoretical analysis or formal guarantees to support this statement. It remains unclear whether this approach eliminates concept drift or merely alleviates it empirically. Moreover, the paper does not analyze model behavior when the test-time definition $Z_{test}$ is semantically distant from those seen during training, which is crucial for validating true open-world generalization.

4. While the introduction of the PreVAD dataset is commendable, the paper does not explain how annotation consistency and hallucination control were ensured when using large language models or human-in-the-loop annotation processes. Without a transparent validation protocol, the reliability and semantic precision of the textual anomaly descriptions remain uncertain.

**Questions:**

1. As pointed out in the weaknesses, in the era of powerful Vision-Language Models, definition drift can often be addressed through prompting and leveraging pretrained multimodal knowledge, as shown in recent training-free approaches such as LAVAD, SUVAD, and VERA. Since the proposed task setup overlaps with the goals of these works, could the authors clarify why these methods were not discussed or compared, and explain what conceptual or empirical advantages LaGoVAD offers over such VLM-based prompt-driven solutions?
2. The paper claims that explicitly modeling the mapping $\Phi :(V,Z)\rightarrow Y$ can effectively avoid concept drift, yet this statement is not theoretically or empirically justified. Could the authors elaborate on whether this modeling truly eliminates concept drift or merely mitigates it in practice, and provide further analysis on how the model behaves when the test-time definition $Z_{test}$ is semantically distant from training-time definitions $Z_{train}$?
3. Regarding the construction of the PreVAD dataset, the paper does not provide sufficient information about how annotation consistency and reliability were maintained, especially if large language models or automated tools were involved. Could the authors clarify what steps were taken to ensure annotation coherence, prevent hallucinated descriptions, and validate the semantic accuracy of the generated textual labels?

---

> ### Author Response · Authors · 2025-11-21
> **Rebuttal for Reviewer yCS1**
>
> **W1 & Q1 (Comparison with VLMs):**
>
> Thanks for raising this concern. We did not intend to avoid discussing VLM-based approaches. We have discussed and compared with representative VLM-based methods in Section 2.2 and Section 6, including HolmesVAU, Follow the Rules, LAVAD and general-purpose VLMs (Qwen-VL series). We provided quantitative metrics in Tables 2 and 4, and introduced the compared methods both in Section 6.1 and Section F.
>
> Adapting anomaly definitions via prompt engineering is indeed a straightforward idea. However, existing methods mainly focus on improving performance under standard evaluation setups rather than explicitly addressing definition drift. In addition, these approaches typically describe the frames/clips first and then predict scores based on the text, which weakens contextual understanding and leads to less reliable scores. Furthermore, current *training-free* methods tend to be *testing-heavy*, requiring multiple computationally expensive forward passes of large models, which complicates deployment.
>
> Experimentally, as shown in Tables 2, 4, we observe that our method consistently outperforms these training-free methods, which demonstrates that the problem of definition drift has not been effectively mitigated. A rough estimate also shows that our evaluation process completes within a few minutes, whereas the training-free alternatives typically require more than ten hours, which highlights the efficiency advantage of our approach.
>
> We have added the above discussions in Section 2.2.
>
>
>
> **W2 (Simple architecture):**
>
> We appreciate this comment. However, our work prioritizes identifying the core challenge of concept drift over designing complex and specialized architectures, and addressing them effectively. As discussed in the Section 1 Lines 73-90, the core challenge of concept drift lies in two aspects:
>
> 1. the ability of incorporating external guidance (language) into the model’s decision process
>
> 2. the risk of overfitting caused by the reduced sample density under a more complex multimodal space.
>
> To address the first issue, we not only condition the prediction on language definitions, but also use descriptions within the batch as variable definitions during training.
>
> To address the second issue, we propose two architecturally independent regularizers, which effectively increase the sample density and can be seamlessly integrated into more sophisticated architectures in the future.
>
>
>
> **W3 & Q2 (Theory of $(V,Z) \rightarrow Y$ and discussion of $Z_{test}$):**
>
> **Theory of $\Phi: (V,Z) \rightarrow Y$**
>
> Our core assumption is that $Y$ is determined solely by $Z$ and $V$; that is, the anomaly decision depends only on the video and the definition of the anomaly. As detailed in the revised Section 3, this assumption leads to $P_{\text{train}}(Y \mid V) \neq P_{\text{test}}(Y \mid V)$ and $P_{\text{train}}(Y \mid V, Z) = P_{\text{test}}(Y \mid V, Z)$, which are proved in Section I. It means that the traditional modeling method ($\Phi: V \rightarrow Y$) is vulnerable when anomaly definition drifts, while our new approach ($\Phi: (V,Z) \rightarrow Y$) is robust.
>
> In other words, in the idealized setting where $\Phi$ is learned perfectly, explicitly modeling $(V,Z) \rightarrow Y$ theoretically eliminates definition-induced concept drift. In practice, however, our model is instantiated as a parametric deep network trained from finite data, so its performance still depends on the architecture and optimization. Empirically, we observe that our implementation substantially mitigates concept drift, as reflected in the improvements under Protocol 2.
>
> To further clarify the scope of this assumption, Section H discusses several special cases.
>
> - One scenario is that additional latent factors beyond the explicit definition influence abnormality, these factors can be absorbed into an augmented definition, after which the assumption remains valid.
>
> - Another scenario is that the anomaly definition is entirely dependent on the video. Yet in practice, many anomaly definitions are not visually inferable from the video alone. For example, when it is impossible to determine from the scene whether smoking is permitted, the abnormality of smoking has to depend on an external definition.
>
> **Discussion of $Z_{test}$**
>
> Both of our evaluation protocols explicitly consider cases where the test-time definition differs from the training-time definition. As shown in Section E.1, the test sets we use contain a large number of unseen categories, leading to substantial semantic gaps between training and testing definitions. Moreover, compared with related works, which typically conduct zero-shot evaluation on only one or two datasets, our evaluation spans 7 test sets with two protocols, providing stronger evidence for open-world generalization.

---

> ### Author Response · Authors · 2025-11-21
> **Rebuttal for Reviewer yCS1**
>
> **W4 & Q3 (PreVAD Annotation):**
>
> Thank you for the question and for recognizing the contribution of PreVAD. We took multiple steps to ensure annotation consistency and to control hallucination.
>
> As described in Section D.2, we used several state-of-the-art models jointly and applied prompt-engineering strategies to clean and annotate the data. For human annotation, all annotators had domain expertise in video anomaly detection, and samples with inconsistent labels underwent a dedicated review process.
>
> For MLLM-generated descriptions, we conducted subjective quality assessment. Three annotators independently evaluated 200 randomly sampled videos using a 1–5 scoring scale, with the criteria showing in Table B. Descriptions generated *without* constrained prompt received an average score of 3.5, whereas those generated *with* constrained prompt achieved an average score of 4.3, demonstrating the effectiveness of the constrained prompt. We also assess the description annotations of HIVAU-70K dataset used by Holmes-VAU, which receive a score of 4.4, indicating the noise level in our annotation is consistent with that of other works. To quantify agreement between annotators, we computed the intraclass correlation coefficient (ICC) [i], obtaining ICC(2,k) = 0.799, which indicates strong inter-annotator consistency. The details of the subjective quality assessment are in Section D.2.
>
> [i] Shrout, P. E., & Fleiss, J. L. (1979). Intraclass correlations: uses in assessing rater reliability. Psychological bulletin, 86(2), 420.

---

> > ### Comment · Reviewer_yCS1 · 2025-11-27
> > **I am basically satisfied with the authors' responses**
> >
> > After reading the authors’ rebuttal, I believe some of my concerns have been addressed. In particular, the response to the second point clarifies the intended design philosophy, and when considered alongside the current development trajectory of LLMs and VLMs, it helps me better understand both the strengths and the limitations of the proposed work.
> >
> > To be frank, I still hold the view that for the problem of varying anomaly definitions across different VAD scenarios, the latest vision-language models (e.g., Qwen3-VL, Seed-Vision, or Gemini 3.0) are already capable of performing zero-shot anomaly redefinition at fairly high speed. Some recent experiments I conducted further support this observation.
> >
> > However, this does not diminish the value of the authors’ approach. Despite the prevailing trend toward larger-scale and higher-quality models and data, a lightweight, trainable, and computationally efficient solution remains meaningful. In practical deployment, different application scenarios involve different trade-offs among model size, inference speed, and computational cost, and in that sense, the proposed method provides practical utility and retains research significance.
> >
> > Additionally, I appreciate the effort the authors put into dataset annotation. I strongly encourage the authors to release the dataset as soon as possible, as large-scale, high-quality datasets are exceptionally valuable resources for the VAD community.
> >
> > Overall, I am willing to raise my score to 8. However, I have to lower my confidence to 3, because I remain uncertain how the community will perceive lightweight training-based approaches in light of the rapid progress of VLMs—whose performance, generalization ability, and computational efficiency are improving to the point that some advanced VLMs can already outperform works like LAVAD via dense sampling over full videos at relatively high FPS.

---

### Author Response · Authors · 2025-11-27

Dear Reviewers,

Thank you very much for taking the time to review our work. We have submitted the revised paper and detailed responses accordingly. Since the review deadline is approaching, if any concerns remain, please let us know. Otherwise, we would appreciate your consideration for increasing the score. Your expert insights will be highly helpful for further improving the work.

Sincerely,

The Authors

---

### Meta-Review · Program_Chairs · 2026-01-07

**Summary:**

The initial ratings are 6, 6, 6, 4. This paper proposes LaGoVAD, a framework for video anomaly detection (VAD) in open-world scenarios, where anomaly definitions can change dynamically according to user-specified natural language input. Also, the authors introduce PreVAD, a dataset with multi-level anomaly categories and textual descriptions.

Strengths:
(1)The paper introduces the PreVAD, a large-scale, multi-domain dataset with rich textual annotations and hierarchical labeling.
(2)Comprehensive evaluation across 7 datasets with a dedicated concept drift protocol.
Weaknesses:
(1)The evaluation under Protocol 2 is incomplete. It only considers concept drift in one direction—where abnormal events in the pre-training dataset (PreVAD) may shift to normal. The reverse scenario, in which normal events become abnormal, is not evaluated. It would be beneficial to include this additional evaluation.
(2)While the two proposed regularization strategies are technically sound, both video splicing and hard negative mining have been extensively explored in prior work.

**This paper is being conditionally accepted provided the authors address the following in their camera-ready**:
[Ethics concerns] Authors should describe the license for the released videos, and acknowledge any legal risks associated with their use and release of the data. They should also either remove PII (such as blurring license plates and faces) or detail the steps they have taken to obtain consent for releasing this data.

**Reviewer Concerns:**

Some concerns of Reviewer yCS1 and 2UdC were addressed by the rebuttal, and Some main concerns of  Reviewer BtGb and vsXK are still outstanding.

**Reviewer Scores:**

Reviewer yCS1 and 2UdC maybe raise the rating score.

---

### Decision · Program_Chairs · 2026-01-26

**Decision:**

Accept (Poster)

**Comment:**

Conditions for acceptance have been satisfied.